# Hyperbolic Hierarchical Clustering for Visual Representation Learning

## Abstract

We investigate the token mixer in vision backbones by revisiting clustering, one of the most classic approaches in machine learning. An effective token mixer is a fundamental component of modern vision backbones like vision Transformers, facilitating information exchange between image patches. Mainstream token mixers, which rely on convolution, attention, MLP, or their hybrids, primarily focus on navigating the trade-off between accuracy and computational cost. However, a significant drawback of these methods is their black-box nature; their encoding process is opaque and lacks interpretability. Diverging from these opaque designs, we introduce *ClusterMixer*, a transparent token mixer that is grounded in a clustering paradigm and interpretable by design. *ClusterMixer* explicitly formulates the token mixing process through a hierarchical clustering mechanism. To model the natural, tree-like relationships inherent in visual data, the clustering is performed in hyperbolic space, which is well-suited for embedding hierarchies with low distortion. Building on this innovation, we present HCFORMER, a new backbone architecture that integrates *ClusterMixer* with a series of meticulously designed clustering strategies to ensure robust performance across tasks. Extensive experiments demonstrate that HCFORMER consistently outperforms its counterparts across diverse tasks, including image classification, object detection, instance segmentation, and semantic segmentation. Considering its transparency and efficacy, we hope HCFORMER can facilitate a paradigm shift toward interpretable backbones. Our source code will be released.

## 1 Introduction

Convolutional Networks (ConvNets) and Vision Transformers (ViTs) are the dominant paradigms in computer vision. ConvNets serve as the *de facto* standard since the pioneering success of AlexNet [2], owing to their strong inductive bias (*i.e.*, locality and translation equivariance). Marking a paradigm shift, ViTs [3] introduce self-attention mechanisms with fewer inductive biases, enabling effective scaling and superior generalization over ConvNets. Building on this, MLP-Mixer [4] conceptualizes *token mixing* and proposes a convolution- and attention-free alternative, which replaces the self-attention layers with spatial MLPs. This work spurs research into alternative token mixers in the vision community. For instance, VAN

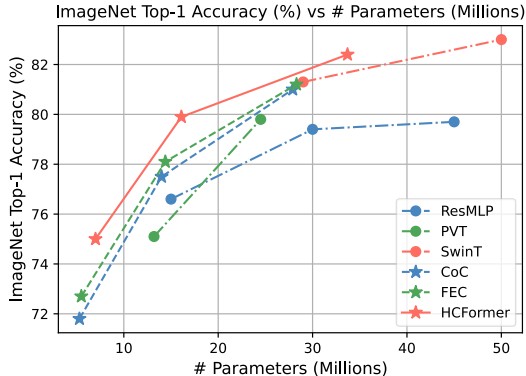

Figure 1: Top-1 accuracy of HCFormer and other state-of-the-art models on ImageNet-1K [1].

[5] employs convolution as the token mixer, while Uniformer [6] integrates both convolution and self-attention. Despite their incremental performance gains, the reasoning process of these models remains opaque for human. Recently, there has been growing research interest in clustering-based vision backbones [7, 8], which can provide enhanced interpretability. Albeit promising, it has not yet fully exploited the capabilities of clustering algorithms for visual representation learning, as evidenced by their suboptimal performance, indicating a notable gap that merits further investigation.

In this work, we introduce an efficient token mixer based on clustering algorithms, dubbed *Cluster-Mixer*, which operates via: **i)** initializing data points and cluster centers based on token representations, **ii)** assigning data points to centers to form distinct clusters, and **iii)** performing token mixing based on the established clusters. Nevertheless, the direct application of *ClusterMixer* presents a dual challenge in terms of computational efficiency and model performance. **First**, it faces analogous challenges of computational complexity as self-attention and spatial MLPs, stemming from data point and cluster center counts. This efficiency limitation becomes more pronounced for vision tasks that require extensive token sets to accomplish dense prediction or high-resolution image representation. A straightforward approach is to reduce the number of cluster centers; however, it leads to an undesirable tradeoff by degrading model performance. **Second**, the similarity estimation of clustering algorithms is primarily conducted in Euclidean geometry [7, 9]. However, the *flat geometry* of Euclidean space limits its ability in modeling hierarchical relation [10], leading to distortions in the semantic distance and suboptimal performance. In other words, even if two embeddings are close in Euclidean geometry, they may be distant in the semantic hierarchy.

To bridge these gaps, we propose HCFORMER, a framework for visual representation learning that enhances *ClusterMixer* via two meticulously designed strategies: **i)** *hierarchical clustering*. Here input patches are partitioned into several non-overlapping windows, with token mixing restricted to local windows to reduce computational complexity in clustering operations. To capture global contextual information, we further apply a mixing operation over these windows, which introduces only minimal computational overhead. **ii)** *hyperbolic hierarchical clustering*. Unlike Euclidean geometry, which excels in flat, simple structures, hyperbolic geometry naturally models hierarchical relationships as a continuous analog of trees [11], thereby better capturing abstract and complex semantics. To leverage these complementary properties, we extend the *ClusterMixer* to use Euclidean space for fine-grained patch-level clustering and hyperbolic space for abstract window-level clustering.

Despite its attractive properties (*e.g.*, grasping abstract and complex semantic relationships), hyperbolic geometry is numerically unstable, whereas Euclidean geometry demonstrates superior performance and stability in simple scenarios. Besides, the operations defined in hyperbolic geometry incur greater computational overhead compared to their Euclidean counterparts. To address these trade-offs, we extend the *ClusterMixer* to perform clustering across dual geometries: Euclidean similarity is computed at the finer-grained and computationally demanding patch-level, while hyperbolic similarity is estimated at the abstract and computationally efficient window-level.

HCFORMER enjoys several desirable virtues: **First**, *efficiency*. Based on the proposed clustering strategies, HCFORMER can reduce the quadratic computational complexity associated with increasing data points and cluster centers to a linear one. This reduction in complexity is especially beneficial for downstream tasks, *e.g.*, semantic segmentation and object detection, which facilitates the use of a greater number of cluster centers for improved performance. **Second**, *holistic hierarchy*. HCFORMER conducts *ClusterMixer* in both Euclidean and Hyperbolic geometries, enabling the model to handle similarity estimation across diverse scenarios, from simple (*e.g.*, flat relation) to complex (*e.g.*, tree-like structure). **Third**, *flexibility*. Owing to its non-parametric and clustering-based design, *ClusterMixer* enables HCFORMER to be effectively applied to various downstream tasks (see §4).

For comprehensive evaluation, HCFORMER is benchmarked on the three datasets covering diverse application scenarios, including ImageNet-1K [1] for image classification, ADE20K [12] for semantic segmentation, COCO [13] for instance detection and segmentation. In §4.1, by training from scratch, HCFORMER outperforms mainstream counterparts with similar parameter counts, *e.g.*, exceeding ResNet-50 by 2.6% and Swin-Tiny by 1.1% in `top-1` accuracy. Notably, substantial improvements (1.0%~3.2%) can also be observed compared with other clustering-based backbones. In §4.2, as the backbone with Semantic FPN [14], HCFORMER achieves performance gains of 0.7%~2.5% mIoU for semantic segmentation. In §4.3, HCFORMER integrated with Mask R-CNN [15] demonstrates competitive performance for instance detection and segmentation. These results are impressive for a clustering-based backbone like HCFORMER, which also enjoys built-in interpretability.

## 2 RELATED WORK

**Clustering in Vision.** The objective of clustering is to partition finite, unlabeled data into discrete subsets of inherent groupings or clusters using a similarity measure (*e.g.*, Euclidean distance) [16, 17], operating as an essential tool in pattern recognition. Traditional clustering approaches are adopted

in computer vision as an effective image preprocessing method [18–20], which groups pixels into perceptually meaningful atomic regions. Modern clustering-based methods are further developed for downstream vision tasks, such as semantic segmentation [21, 22] and trajectory prediction [23, 24]. Due to its flexibility and interpretability, clustering methods are increasingly utilized for modeling complex data types, *e.g.*, point clouds [25, 26] and protein [27, 28], and employed to elucidate the decisions derived from the neural network [29, 30]. Motivated by these compelling virtues, our work endeavors to explore the adaptation of clustering methods for basic visual representation learning.

**Learning in Hyperbolic Geometry.** Hyperbolic geometry has attracted significant attention owing to its effectiveness in modeling data with tree-like structures [31, 32]. These desirable properties render hyperbolic embeddings a superior alternative in diverse modalities, *e.g.*, graphs [33, 34], images [35, 36] and videos [37, 38]. Another important line of research involves leveraging hyperbolic geometry for representation learning. For instance, Hyperbolic Neural Networks [39] introduce a set of corresponding hyperbolic neural network layers and demonstrates superior performance over their Euclidean counterparts in various downstream tasks, such as text entailment and noisy prefix prediction. Subsequent works further expand this framework: Hyperbolic Neural Networks++ [40] proposes hyperbolic convolutional layers, while Hyperbolic attention networks [41] present a Graph Neural Network architecture that operates in hyperbolic space. In contrast to these works, this paper incorporates hyperbolic geometry into clustering algorithms for visual representation learning, aiming to enhance token mixing by better capturing hierarchical relationships between features.

**Generic Vision Backbone.** Revolutions in computer vision can be characterized as a paradigm shift in feature extraction. In early research [42–44], feature extraction typically relies on manually crafted features (*e.g.*, geometric structure or color statistics) derived from the predefined rules. Beginning with AlexNet [2], Convolutional Networks (ConvNets) [45, 46] evolve this paradigm from hand-crafted feature engineering to data-driven feature learning, which offers greater versatility and performance. ConvNets utilize sliding windows to partition the entire image into a set of rectangular patches, where convolutional kernels conduct pixel mixing via weighted summations in the local region. In contrast, Vision Transformers (ViTs) [3] provide an attention-based alternative, which reduces the image-specific inductive biases in ConvNets for feature extraction. ViTs split images into a collection of non-overlapping patches and conduct token mixing across them in a global range, facilitating model scalability. Beyond these two paradigms, Context Cluster (CoC) [7] proposes an innovative idea to group the points into clusters, where pixel features are aggregated and then dispatched within a cluster. This simple design is convolution- and attention-free, which only relies on a clustering algorithm to provide interpretability. FEC [8] extends this by introducing clustering-based feature pooling for downsampling, thereby achieving a fully clustering-based feature extraction.

Nevertheless, the potential of clustering methodologies for representation learning remains under-explored within the research community. Our work achieves strong performance on various vision tasks, and we hope it will contribute to a paradigm shift toward clustering-based vision backbones.

## 3 METHOD

### 3.1 OVERALL ARCHITECTURE

An overview of the HCFORMER is presented in Figure 2, which is built upon the MetaFormer paradigm [47]. The input image **I** is first split into non-overlapping patches, following the patch splitting module of ViT [3]. Given the fundamental role of spatial proximity in visual clustering [48], these patches are equipped with their coordinates and processed into embedding tokens. Then, a series of residual blocks incorporating token mixers are applied on these tokens. Instead of attention mechanism (e.g., Transformer [49]), spatial MLP (e.g., MLP-Mixer [4]), or average pooling (e.g., PoolFormer [47]), we implement interpretable clustering algorithms as the token mixer, dubbed *ClusterMixer*. To produce a hierarchical representation akin to ConvNets [45, 46], the number of tokens is reduced by concatenating the spatially adjacent tokens, which is achieved by a convolutional operation. This design enables seamless adaptation to downstream tasks such as semantic segmentation and object detection. The following section first delineates the implementation of *ClusterMixer* (§3.2) and then describes the two strategies developed to enhance it (§3.3 & §3.4).

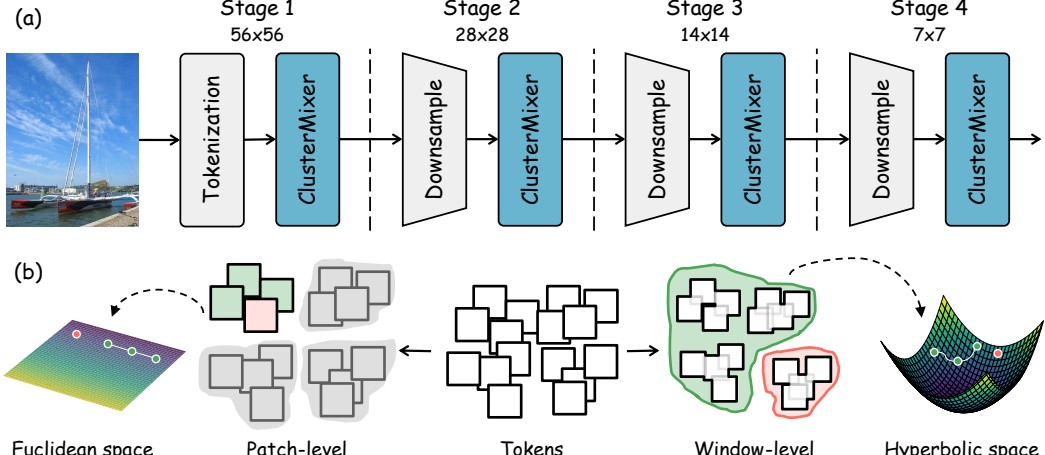

Figure 2: (a) **Overall pipeline of HCFORMER.** Following the MetaFormer paradigm [47], HC-FORMER adopts a hierarchical architecture with 4 stages. (b) **Hierarchical clustering**. *ClusterMixer* conducts patch-level clustering in Euclidean space and window-level clustering in hyperbolic space.

## 3.2 CLUSTERMIXER

Considering data points $\mathbf{X} \in \mathbb{R}^{N \times C}$, the *ClusterMixer* is performed as following:

**Center Estimation.** The estimation of cluster centers from data points can be implemented through various methods, such as the K-means [18] or the Sinkhorn-Knopp algorithm [30, 50]. However, these methods are computationally intensive and thus not suitable for token mixing. In this paper, we initialize the data points $\mathbf{X} \in \mathbb{R}^{N \times C}$ with feature embeddings and employ average pooling over neighboring patches to estimate cluster centers $\mathbf{C} \in \mathbb{R}^{M \times C}$, inspired by [47]. This approach incurs a computational complexity linear in the sequence length while involving no learnable parameters.

**Cluster Assignment.** We first partition all data points into distinct clusters by assigning them to different cluster centers $c_j \in \mathbf{C}$ based on the feature similarity. Notably, in contrast to hard clustering (*i.e.*, each token is exclusively assigned to a single cluster), we employ a soft clustering approach, where data points can belong to multiple clusters with varying degrees of probability. This process resembles the Expectation-step (E-step) in Expectation-Maximization (EM) clustering [51], with the distinction that our cluster centers are precomputed via average pooling over neighboring patches, avoiding iterative updates. The $i$-th row $a_i \in \mathbb{R}^{1 \times M}$ of the assignment matrix $\mathbf{A} \in \mathbb{R}^{N \times M}$ for each token $x_i \in \mathbf{X}$ is computed as:

$$a_i = \text{Softmax}_{c_j \in \mathbf{C}, b_{i,j} \in \mathbf{B}}(\alpha \cdot D(x_i, c_j) + b_{i,j}), \tag{1}$$

where $D$ denotes the similarity metric (see §3.4), $\alpha$ is a learnable scalar for feature similarity, and $\mathbf{B} \in \mathbb{R}^{N \times M}$ parameterizes the learnable relative positional bias.

**Token Mixing.** We perform token mixing in two steps: **i)** feature aggregation from data points and **ii)** feature propagation back to data points. Specifically, it adaptively gathers contextual information from data points based on the assignment matrix to compute the aggregated feature $g_i \in \mathbb{R}^{N \times 1}$:

$$g_i = \sum_{j=1}^{M} \mathbf{A}_{i,j} \cdot c'_j, \quad s.t. \;\; c'_j = (c_j + \sum_{i=1}^{N} \mathbf{A}_{i,j} \cdot x_i)/(1 + \sum_{i=1}^{N} \mathbf{A}_{i,j}). \tag{2}$$

The aggregated feature $g_i$ is then employed to update the corresponding data point via:

$$x'_i = x_i + \text{FC}(g_i), \tag{3}$$

where FC denotes a fully-connected layer to maintain the feature dimension of token embeddings.

Our *tokenMixer* offers several advantages: **i)** the soft clustering (*vs* CoC's hard clustering [7]) enables our method to benefit from constructing more clusters across various tasks (see §4.4). **ii)** the relative positional bias effectively assists the clustering algorithm in establishing relational dependencies.

### 3.3 Hierarchical Clustering

To tackle the challenge of computational complexity, we perform clustering operations within non-overlapping local windows, where each input is partitioned into $S$ windows, each containing $K$ patches. However, such disconnected windows are recognized to limit the acquisition of global contextual information for representation learning. While this limitation can be mitigated via shifted windows in Swin Transformer [52], our clustering-based token mixer offers an alternative approach by performing mixing at both the *patch-level* and the *window-level*. These two strategies differ solely in *how data points and cluster centers are formulated* in the *ClusterMixer*:

- *Patch-level clustering*. Each patch serves as a data point, while cluster centers are estimated within each localized window, and the *ClusterMixer* is performed in parallel across all windows.
- *Window-level clustering*. Each window is modeled as a data point, represented by the mean feature embedding of its patches, while cluster centers are estimated globally over the entire input.

This formulation transforms the quadratic complexity of *ClusterMixer* into linear complexity. Empirically, this hierarchical clustering mechanism is better suited for the proposed *ClusterMixer* compared to the shifted windowing scheme (see §4.4).

### 3.4 Hyperbolic Hierarchical Clustering

The choice of similarity metric in Equation 1 is critical for cluster assignment. Currently, similarity estimation is primarily conducted in Euclidean geometry [7, 9]: given data points $x_{\mathbb{E}} \in \mathbb{R}^C$ and a cluster center $c_{\mathbb{E}} \in \mathbb{R}^C$, the feature distance is estimated by pair-wise cosine similarity:

$$D_{\mathbb{E}}(x_{\mathbb{E}}, c_{\mathbb{E}}) := \text{sim}(x_{\mathbb{E}}, c_{\mathbb{E}}) = \frac{x_{\mathbb{E}} \cdot c_{\mathbb{E}}}{\|x_{\mathbb{E}}\|\|c_{\mathbb{E}}\|}, \tag{4}$$

where $\|\cdot\|$ denotes the L2 norm. To better model abstract and complex semantic relationships, we next describe how to perform similarity estimation in hyperbolic space.

**Defining Hyperboloid.** Hyperbolic spaces are Riemannian manifolds characterized by a constant negative curvature, whereas Euclidean spaces exhibit zero-curvature (*i.e.*, flat) geometry. There are five well-known isometric models of hyperbolic geometry, including the Lorentz model, the Poincaré ball model, the Poincaré half space model, the Klein model, and the Hemishpere model [53]. Among these, the Lorentz model is widely adopted due to its numerical stability and computational efficiency.

The Lorentz model is an $n$-dimensional hyperbolic space on the upper half of a two-sheeted hyperboloid in $n+1$-dimensional Minkowski space. In the Lorentz space, every vector $x \in \mathbb{R}^{n+1}$ can be written as $[x_{\text{time}}, x_{\text{space}}]$, where $x_{\text{time}} \in \mathbb{R}$ denotes the first dimension as the *time dimension* and $x_{\text{space}} \in \mathbb{R}^n$ denotes the remaining $n$ dimensions as the *space dimension*. The model is described as:

$$\mathbb{L}^n = \{x \in \mathbb{R}^{n+1} : \langle x, x \rangle_{\mathbb{L}} = -1/\kappa, \kappa > 0\}, \quad s.t. \ x_{\text{time}} = \sqrt{1/\kappa + \langle x_{\text{space}} \rangle^2}, \tag{5}$$

where $-\kappa \in \mathbb{R}$ is the curvature of the space, typically set to $\kappa = 1$ for simplicity, and $\langle \cdot, \cdot \rangle_{\mathbb{L}}$ denotes the *Lorentzian inner product*, defined as:

$$\langle x, y \rangle_{\mathbb{L}} := -x_{\text{time}} y_{\text{time}} + x_{\text{space}}^T y_{\text{space}}. \tag{6}$$

**Lifting onto Hyperboloid.** To project a vector from Euclidean space to hyperbolic space, we first define a mapping from the *tangent space* $T_z\mathbb{L}^n$ onto the *hyperbolic manifold* $\mathbb{L}^n$, where $T_z\mathbb{L}^n$ is a Euclidean space of vectors that are orthogonal to some points $z \in \mathbb{L}^n$ on the hyperboloid. Given a tangent vector $v \in T_z\mathbb{L}^n$, the *exponential mapping* $T_z\mathbb{L}^n \to \mathbb{L}^n$ can be defined as:

$$\text{expm}_z^\kappa(v) = \cosh(\sqrt{\kappa}\|v\|_{\mathbb{L}})z + \sinh((\sqrt{\kappa}\|v\|_{\mathbb{L}})\frac{v}{\sqrt{\kappa}\|v\|_{\mathbb{L}}}, \quad s.t. \ \|v\|_{\mathbb{L}} = \sqrt{\langle v, v \rangle_{\mathbb{L}}}, \tag{7}$$

By treating Euclidean vectors as tangent vectors at the origin $\mathbf{0} = (\sqrt{1/\kappa}, 0, \ldots, 0)^T$ of hyperbolic space, one can map vectors from Euclidean space to hyperbolic space via $\text{expm}_0(\cdot)$.

Given a data point $x_{\mathbb{E}} \in \mathbb{R}^C$ and a cluster center $c_{\mathbb{E}} \in \mathbb{R}^C$, we first project both onto the Lorentz hyperboloid $\mathbb{L}^n$, yielding embeddings $x_{\mathbb{L}} \in \mathbb{L}^n$ and $c_{\mathbb{L}} \in \mathbb{L}^n$ in the hyperbolic space:

$$x_{\mathbb{L}} = \text{expm}_0^\kappa(x_{\mathbb{E}}), \quad c_{\mathbb{L}} = \text{expm}_0^\kappa(c_{\mathbb{E}}). \tag{8}$$

**Estimating Hyperbolic Similarity.** Given two hyperbolic embeddings $x_{\mathbb{L}}, c_{\mathbb{L}} \in \mathbb{L}^n$, we estimate hyperbolic similarity via the Lorentzian distance, as:

$$D_{\mathbb{L}}^{\kappa}(x_{\mathbb{L}}, c_{\mathbb{L}}) := \sqrt{1/\kappa} \cdot \cosh^{-1}(-\langle x_{\mathbb{L}}, c_{\mathbb{L}} \rangle_{\mathbb{L}}). \tag{9}$$

**Reformulation of ClusterMixer.** Despite its attractive properties (*e.g.*, grasping abstract and complex semantic relationships), hyperbolic geometry is numerically unstable, whereas Euclidean geometry demonstrates superior performance and stability in simple scenarios. Besides, the operations defined in hyperbolic geometry incur greater computational overhead compared to their Euclidean counterparts. To address these trade-offs, we extend the *ClusterMixer* to perform clustering across dual geometries: Euclidean similarity is computed at the finer-grained and computationally demanding patch-level, while hyperbolic similarity is estimated at the abstract and computationally efficient window-level. Drawing upon this principle, Equation 3 is rewritten as:

$$x' = x + \text{FC}(\text{Norm}([g_{\mathbb{L}}^W, g_{\mathbb{E}}^P])), \tag{10}$$

where $[\cdot, \cdot]$ denotes the concatenation operation. Here, $g_{\mathbb{L}}^W$ represents features aggregated from windows via Hyperbolic similarity, while $g_{\mathbb{E}}^P$ does so for patches using Euclidean similarity. The computations for both are performed in parallel and remain decoupled from one another.

### 3.5 NETWORK CONFIGURATION

Following conventional protocols [3, 47], the network input for image classification is set to $224 \times 224$, yielding $224/4 \times 224/4 = 56 \times 56$ patches. The network architecture comprises four stages, with the number of patches reduced by a factor of four, while each cluster center remains derived from 4 data points throughout the forward process. In this way, global patch-level mixing would impose a significant computational burden on the clustering algorithm. For instance, at stage 1, there would be $56/2 \times 56/2$ cluster centers and $56 \times 56$ data points, resulting in a $784 \times 3136$ assignment matrix. Given $S = 49$ partitioned windows at this stage, our hyperbolic hierarchical clustering strategy reduces the patch-level complexity to $49 \times 16 \times 64$, at the cost of an additional window-level complexity $16 \times 49$. For downstream tasks such as segmentation and detection with variable-sized inputs, reflect padding is applied to maintain this configuration. Additionally, the relative positional bias is linearly interpolated before use to accommodate variable input sizes. See Appendix §B for further details.

## 4 EXPERIMENT

### 4.1 IMAGE CLASSIFICATION

**Dataset.** The evaluation for image classification is carried out on ImageNet-1K [1], which contains 1.3M training images and 50K validation images from common 1K classes.

**Setup.** We adopt `Timm` as the codebase and the experiments are run on 4 A100 GPUs with a batch size of 256. Following [7, 47], all of our models are trained for 310 epochs using a momentum of 0.9 and a weight decay of 0.05. The learning rate is set to 1e-3 and adjusted via a cosine schedule with 5 warmup epochs. For data augmentation, we use Mixup [62], CutMix [63], CutOut [64], and RandAugment [65]. `Top-1` classification accuracy is reported.

**Results.** Table 1 compares HCFORMER with other widely-used baselines on image classification. HCFORMER achieves superior results over counterparts with similar parameter counts, demonstrating the effectiveness of the proposed method. For example, with 16M parameters, HCFORMER outperforms ResMLP-12 by 3.4%, PVT-Tiny by 4.9%, and ConvMixer-1024/12 by 2.2% in `top-1` accuracy. HCFORMER also demonstrates superior performance compared to other models adhering to the MetaFormer paradigm, as evidenced by HCFORMER-Medium outperforming Swin-Tiny (82.4% *vs*. 81.3%) and MLP-Mixer-B/16 (82.4% *vs*. 76.4%). These results are impressive, considering the transparent, clustering-based interpretable nature of HCFORMER. With a marginal increase in parameters and FLOPs, HCFORMER yields a notable improvement compared to existing cluster-based approaches; for instance, it achieves 2.3%/1.8%/1.2% gains in `top-1` accuracy across three configurations of model size relative to FEC [8]. To further substantiate the performance of our method, we introduce a smaller variant of HCFORMER with only 5.1M parameters, *i.e.*, HCFORMER-Nano, which has the fewest parameters among all methods listed in Table 1. Despite its compact size,

Table 1: Classification `top-1` accuracy on ImageNet-1k [1] `val`. Throughput (images/s) measured on a single V100 GPU at batch size 128, averaged over last 500 iterations.

| | Method | | Param. (M) | GFLOPs (G) | Top-1(%) ↑ | Throughputs (images/s) |
|---|---|---|---|---|---|---|
| CLIP | MERU-S/16 | [54][ICML23] | - | - | 34.3 | - |
| | MERU-B/16 | [54][ICML23] | - | - | 37.5 | - |
| | MERU-L/16 | [54][ICML23] | - | - | 38.8 | - |
| | HyCoCLIP-S/16 | [55][ICLR24] | - | - | 41.7 | - |
| | HyCoCLIP-B/16 | [55][ICLR24] | - | - | 45.8 | - |
| | HCL | [56][CVPR23] | - | - | 58.5 | - |
| MLP | ResMLP-12 | [57][TPAMI22] | 15.0 | 3.0 | 76.6 | - |
| | ResMLP-24 | [57][TPAMI22] | 30.0 | 6.0 | 79.4 | - |
| | ResMLP-36 | [57][TPAMI22] | 45.0 | 8.9 | 79.7 | - |
| | MLP-Mixer-B/16 | [4][NeurIPS21] | 59.0 | 12.7 | 76.4 | - |
| | MLP-Mixer-L/16 | [4][NeurIPS21] | 207.0 | 44.8 | 71.8 | - |
| | gMLP-Ti | [58][NeurIPS21] | 6.0 | 1.4 | 72.3 | - |
| | gMLP-S | [58][NeurIPS21] | 20.0 | 4.5 | 79.6 | - |
| Attention | ViT-B/16 | [3][ICLR20] | 86.0 | 55.5 | 77.9 | - |
| | ViT-L/16 | [3][ICLR20] | 307 | 190.7 | 76.5 | - |
| | PVT-Tiny | [59][ICCV21] | 13.2 | 1.9 | 75.1 | - |
| | PVT-Small | [59][ICCV21] | 24.5 | 3.8 | 79.8 | - |
| | DeiT-Tiny/16 | [60][ICML21] | 5.7 | 1.3 | 72.2 | - |
| | DeiT-Small/16 | [60][ICML21] | 22.1 | 4.6 | 79.8 | - |
| | Swin-Tiny | [52][ICCV21] | 29 | 4.5 | 81.3 | - |
| | Swin-Small | [52][ICCV21] | 50 | 8.7 | 83.0 | - |
| Conv. | ResNet-18 | [46][CVPR16] | 12 | 1.8 | 69.8 | - |
| | ResNet-50 | [46][CVPR16] | 26 | 4.1 | 79.8 | - |
| | ConvMixer-512/16 | [61][TMLR23] | 5.4 | - | 73.8 | - |
| | ConvMixer-1024/12 | [61][TMLR23] | 14.6 | - | 77.8 | - |
| | ConvMixer-768/32 | [61][TMLR23] | 21.1 | - | 80.2 | - |
| Cluster | CoC-Tiny | [7][ICLR23] | 5.3 | 1.0 | 71.8 | 792.5 |
| | CoC-Small | [7][ICLR23] | 14.0 | 2.6 | 77.5 | 581.8 |
| | CoC-Medium | [7][ICLR23] | 27.9 | 5.5 | 81.0 | 473.8 |
| | FEC-Small | [8][CVPR24] | 5.5 | 1.4 | 72.7 | 742.1 |
| | FEC-Base | [8][CVPR24] | 14.4 | 3.4 | 78.1 | 532.1 |
| | FEC-Large | [8][CVPR24] | 28.3 | 6.5 | 81.2 | 478.2 |
| | HCFormer-Nano | (ours) | 5.1 | 0.9 | 73.1 | 719.0 |
| | HCFormer-Tiny | (ours) | 7.0 | 1.0 | 75.0 | 536.0 |
| | HCFormer-Small | (ours) | 16.1 | 2.9 | 79.9 | 324.7 |
| | HCFormer-Medium | (ours) | 33.7 | 6.4 | 82.4 | 235.7 |

HCFORMER-Nano achieves 73.1% `top-1` accuracy, exceeding CoC-Tiny by 1.3% with 0.2M fewer parameters and FEC-Small by 0.4% with 0.4M fewer parameters. Notably, our 16M model exhibits performance comparable to ConvMixer-768/32 [61] (21.1M) and DeiT-Small/16 [60] (22.1M). In conclusion, these promising results manifest the effectiveness and wide benefit of our algorithm.

## 4.2 SEMANTIC SEGMENTATION

**Dataset.** The evaluation for semantic segmentation is carried out on ADE20K [12], which includes 20K and 2K images in the training and validation set from 150 semantic categories.

**Setup.** We adopt `mmsegmentation` as the codebase and the experiments are run on 4 A100 GPUs with a batch size of 16. HCFormer serves as the backbone equipped with Semantic FPN [14] for semantic segmentation. For a fair comparison, all models are trained for 80k iterations using AdamW. The learning rate starts at 2e-4, with polynomial decay (power=0.9). The backbones are initialized with ImageNet pre-trained weights and the added layers employ Xavier initialization. During training, we use random scale jittering with a factor in $[0.5, 2.0]$ and a crop size of $512 \times 512$ for training. During inference, we use one input image scale with shorter side as 512 pixels. Mean intersection-over-union (mIoU) is reported.

Table 2: Segmentation `mIoU` score of different backbones with Semantic FPN [14] on ADE20K [12] `val`.

| Backbone | | Params | mIoU(%) ↑ |
|---|---|---|---|
| ResNet-18 | [46][CVPR16] | 15.5M | 32.9 |
| ResNet-50 | [46][CVPR16] | 28.5M | 36.7 |
| PVT-Tiny | [59][ICCV21] | 17.0M | 35.7 |
| PVT-Small | [59][ICCV21] | 28.2M | 39.8 |
| CoC-Small/4 | [7][ICLR23] | 17.6M | 36.6 |
| CoC-Medium/4 | [7][ICLR23] | 25.2M | 40.2 |
| FEC-Small | [8][CVPR24] | 9.1M | 35.3 |
| FEC-Base | [8][CVPR24] | 18.0M | 37.7 |
| FEC-Large | [8][CVPR24] | 31.9M | 40.5 |
| HCFormer-Nano | (ours) | 8.8M | 36.8 |
| HCFormer-Tiny | (ours) | 10.3M | 37.3 |
| HCFormer-Small | (ours) | 18.9M | 40.4 |
| HCFormer-Medium | (ours) | 35.7M | 43.3 |

Table 3: Object detection and instance segmentation results using Mask-RCNN [15] on COCO [13] val2017. $AP^{box}$ and $AP^{mask}$ denote bounding box AP and mask AP, respectively.

| Method | | Param. | $AP^{box}$ | $AP^{box}_{50}$ | $AP^{box}_{75}$ | $AP^{mask}$ | $AP^{mask}_{50}$ | $AP^{mask}_{75}$ |
|---|---|---|---|---|---|---|---|---|
| ResNet-18 | [46][CVPR16] | 31.2M | 34.0 | 54.0 | 36.7 | 31.2 | 51.0 | 32.7 |
| ResNet-50 | [46][CVPR16] | 44.2M | 38.0 | 58.6 | 41.4 | 34.4 | 55.1 | 36.7 |
| PVT-Tiny | [59][ICCV21] | 32.9M | 36.7 | 59.2 | 39.3 | 35.1 | 56.7 | 37.3 |
| PVT-Small | [59][ICCV21] | 44.1M | 40.4 | 62.9 | 43.8 | 37.8 | 60.1 | 40.3 |
| CoC-Small/4 | [7][ICLR23] | 33.6M | 35.9 | 58.3 | 38.3 | 33.8 | 55.3 | 35.8 |
| CoC-Small/25 | [7][ICLR23] | 33.6M | 37.5 | 60.1 | 40.0 | 35.4 | 57.1 | 37.9 |
| CoC-Small/49 | [7][ICLR23] | 33.6M | 37.2 | 59.8 | 39.7 | 34.9 | 56.7 | 37.0 |
| CoC-Medium/4 | [7][ICLR23] | 42.1M | 38.6 | 61.1 | 41.5 | 36.1 | 58.2 | 38.0 |
| CoC-Medium/25 | [7][ICLR23] | 42.1M | 40.1 | 62.8 | 43.6 | 37.4 | 59.9 | 40.0 |
| CoC-Medium/49 | [7][ICLR23] | 42.1M | 40.6 | 63.3 | 43.9 | 37.6 | 60.1 | 39.9 |
| FEC-Small | [8][CVPR24] | 24.3M | 35.6 | 57.5 | 38.2 | 33.6 | 54.7 | 35.7 |
| FEC-Base | [8][CVPR24] | 33.1M | 37.9 | 60.1 | 40.8 | 35.5 | 57.5 | 37.8 |
| FEC-Large | [8][CVPR24] | 47.1M | 39.9 | 62.5 | 43.2 | 37.3 | 59.5 | 39.5 |
| HCFormer-Nano | (ours) | 24.0M | 36.0 | 57.8 | 38.4 | 34.0 | 55.1 | 36.0 |
| HCFormer-Tiny | (ours) | 25.5M | 36.4 | 58.5 | 38.6 | 34.5 | 55.8 | 36.5 |
| HCFormer-Small | (ours) | 34.1M | 38.7 | 60.9 | 41.8 | 36.0 | 58.0 | 38.3 |
| HCFormer-Medium | (ours) | 50.8M | 40.9 | 63.0 | 44.0 | 37.6 | 59.8 | 40.0 |

**Results.** Table 2 illustrates our compelling results over semantic segmentation. In terms of mIoU, our HCFORMER exceeds CoC [7] and FEC [8] by significant improvements with comparable parameter counts: 40.4% *vs*.36.6% *vs*.37.7% at 19M parameters, 43.3% *vs*.40.2% *vs*.40.5% at 35M parameters. Notably, HCFORMER-Small achieves performance comparable to CoC-Medium/4 and FEC-Large, with 6.3M and 13.0M fewer parameters, respectively. Furthermore, our HCFORMER-Nano outperforms FEC-Small by 1.5% with 0.3M fewer parameters, while even surpassing CoC-Small (17.6M parameters). These benchmarking results are significant, which provide solid evidence that our method serves as an effective backbone architecture for semantic segmentation. We attribute this advancement to the adopted soft clustering mechanism and hierarchical clustering strategies, which enable HCFORMER to capture multi-scale features essential for semantic segmentation.

### 4.3 OBJECT DETECTION AND INSTANCE SEGMENTATION

**Dataset.** The evaluation for object detection and instance segmentation is carried out on MS COCO 2017 benchmark [13], which contains 118K training and 5K validation images.

**Setup.** We adopt mmdetection as the codebase and the experiments are run on 4 A100 GPUs with a batch size of 16. HCFormer is adopted as the backbone of Mask R-CNN [15] for both object detection and instance segmentation tasks. The backbones are initialized with ImageNet pre-trained weights and the added layers employ Xavier initialization. All models are trained for 12 epochs ($1\times$ schedule) using AdamW optimizer with an initial learning rate of 1e-4. During training, images are resized with the shorter side at 800 pixels and the longer side $\leq$ 1,333 pixels. During inference, the shorter side is also scaled to 800 pixels. Mean Average Precision (mAP) is adopted for evaluation.

**Results.** Table 3 reports the numerical results for both object detection and instance segmentation. Empirically, our method achieves superior performance over other competitors under identical network sizes across all evaluation metrics. For instance, at the 34M model scale, HCFORMER outperforms recent clustering-based advancements (*i.e.*, CoC [7] and FEC [8]), achieving 38.7% *vs*.37.2% *vs*.37.9% $AP^{box}$ for object detection and 36.0% *vs*.35.4% *vs*.35.5% $AP^{mask}$ for instance segmentation, respectively. With fewer parameters, HCFORMER-Nano outperforms FEC-Small by 0.4% $AP^{box}$ and 0.4% $AP^{mask}$. Moreover, HCFORMER-Medium achieves 40.9% $AP^{box}$ in object detection and 37.6% $AP^{mask}$ in instance segmentation, surpassing all other methods except PVT-Small, to which it is only marginally inferior (by 0.1% $AP^{mask}$) in instance segmentation. However, its advantage narrows when compared to CoC-Medium/49. We posit that this is because object detection performance benefits from the number of cluster centers, which enables CoC-Medium/49 to achieve results competitive with ours. Crucially, this accuracy comes at the expense of efficiency; HCFormer-Medium achieves a 37% higher throughput (14.0 vs. 10.2) than CoC-Medium/49 (as shown in Supplementary Table 7), underscoring its superior computational economy.

Table 4: A set of ablative experiments on ImageNet [66] val. *Hier. Clus.* indicates hierarchical clustering; *Shift.* implies shifted windows; *Hyp. Geo.* represents hyperbolic geometry; *Rel. Pos.* denotes relative position; *Euc.* and *Hyp.* are Euclidean and hyperbolic geometry, respectively.

| Strategy | top-1(%)↑ | top-5(%)↑ |
|---|---|---|
| *Shift.* | 72.7 | 91.1 |
| *Hier. Clus.* | 73.4 | 91.4 |

(a) Hierarchical Clustering for *clusterMixer*.

| Window | Patch | top-1(%)↑ | top-5(%)↑ |
|---|---|---|---|
| *Euc.* | *Euc.* | 73.4 | 91.4 |
| *Hyp.* | *Hyp.* | 74.7 | 92.3 |
| *Hyp.* | *Euc.* | 75.0 | 92.5 |

(b) Geometry for Clustering.

| Hier. Clus. | Hyp. Geo. | Rel. Pos. | top-1(%)↑ | top-5(%)↑ | mIoU(%) ↑ | AP$^{box}$ | AP$^{mask}$ |
|---|---|---|---|---|---|---|---|
|  |  |  | 71.9 | 90.8 | 35.1 | 34.3 | 32.7 |
|  | ✓ | ✓ | 72.7 | 91.1 | 34.6 | 34.4 | 32.8 |
| ✓ |  | ✓ | 73.4 | 91.4 | 35.2 | 34.7 | 33.1 |
| ✓ | ✓ |  | 74.4 | 92.2 | 36.4 | 35.9 | 34.2 |
| ✓ | ✓ | ✓ | 75.0 | 92.5 | 37.3 | 36.4 | 34.5 |

(c) Key Component Analysis.

## 4.4 DIAGNOSTIC EXPERIMENT

For thorough evaluation, we perform a series of ablative studies on ImageNet-1K [1] val2017 for image classification to investigate the following aspects. HCFormer-Tiny is utilized as the baseline.

**Shifted Windows or Hierarchical Clustering.** We first investigate the effectiveness of the proposed hierarchical clustering strategies for acquiring global contextual information in representation learning, compared with the shifted window mechanism [52]. As outlined in Table 4a, while the shifted window mechanism remains effective, our hierarchical clustering strategies are better suited for *ClusterMixer* (*i.e.*, 72.7% → 73.4%). This may arise because the shifted operation only enables localized interaction between adjacent windows, thereby remaining constrained by the limited receptive field.

**Hyperbolic or Euclidean Distance.** We next examine the impact of hyperbolic versus Euclidean geometry on window-level clustering. As summarized in Table 4b, we find that the use of hyperbolic geometry yields a notable performance gain over Euclidean geometry by 1.6% top-1 accuracy. When clustering is performed entirely in hyperbolic space, a marginal performance degradation (0.3% top-1) is observed, accompanied by a substantial reduction in computational efficiency (throughput decreases from 536.0 to 319.2).

**Key Component Analysis.** We finally ablate the key design elements in the proposed HCFORMER. As shown in Table 4c, the baseline of our model achieves only 71.9% without the proposed components. Empirical results demonstrate that all components provide complementary benefits, as the absence of either leads to performance degradation: -2.3% without hierarchical clustering, -1.6% without hyperbolic geometry, and -0.6% without relative position.

## 4.5 VISUALIZATION OF CLUSTERING

Our configuration ensures that the feature representation at each center of the network architecture (except the final stage) is derived from a $2 \times 2$ token grid, while tokens are progressively merged to reduce their count by a factor of $2 \times 2$ by downsampling operator (See Appendix §B). This mechanism indicates that HCFormer develops gradually expanding clusters during feature extraction, ultimately forming 16 clusters at the final stage. Follow FEC's visualization protocol [8], we also use K-Means to reduce the number of clusters for better visualization. As illustrated in Figure 3, the clustering results demonstrate that our HCFormer is able to capture intrinsic relational patterns among tokens.

## 5 CONCLUSION

Clustering represents a promising yet underexplored direction in architectural design, lacking an effective and efficient framework that enables it to emerge as a competitive alternative to mainstream architectures. This paper proposes a new token-mixer design, termed *ClusterMixer*, which leverages clustering algorithms to enhance token aggregation. To leverage this design effectively, we advocate a universal vision framework, designated as HCFORMER, along with a set of training strategies tailored for *ClusterMixer*. Empirical results demonstrate that this framework improves interpretability over

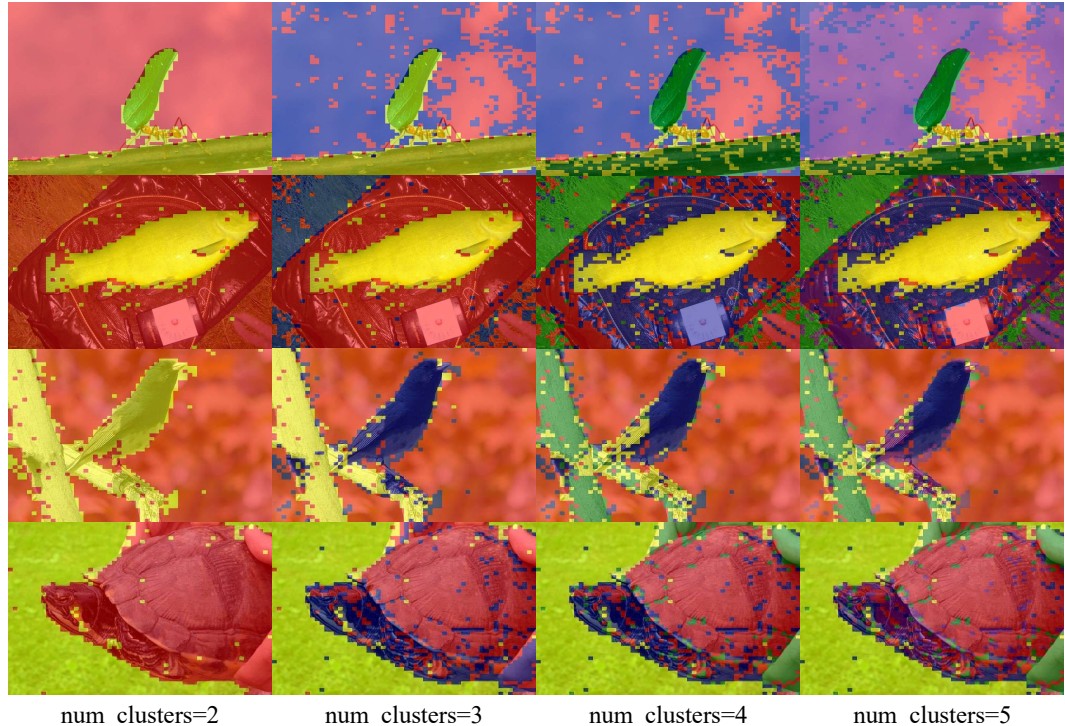

num_clusters=2  num_clusters=3  num_clusters=4  num_clusters=5

Figure 3: Visualization of clustering maps for our HCFormer-Tiny on ImageNet [66] `val`. Different colored masks indicate different clusters, ranging from 2 to 5.

conventional convolution- and attention-based methods while achieving superior performance to existing cluster-based approaches. Given its favorable balance of interpretability and performance, we expect that this approach will potentially benefit a wider range of visual tasks.

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

## SUMMARY OF THE APPENDIX

This appendix contains additional details for the ICLR 2026 submission, titled *"Hierarchical Clustering for Visual Representation Learning"*. The appendix is organized as follows:

- §A provides the pseudo code of HCFORMER.
- §B presents more implementation details of HCFORMER.
- §C gives more diagnostic experiments of HCFORMER.
- §D demonstrates more efficiency analysis of HCFORMER.
- §E introduces the detailed model configurations of HCFORMER.
- §F discusses our limitations, broader impact, and directions of our future work.
- §G provides the Large Language Models (LLMs) usage statement in this work.

## A    PSEUDO CODE

The pseudo code for the instantiation of *TokenMixer* in HCFORMER is given in Algorithm 1. Moreover, to guarantee reproducibility, our full code and pre-trained models will be publicly available.

---

**Algorithm 1** Pseudo-code for *TokenMixer* in a PyTorch-like style.

---

```
"""
X: input patches.
"""
def TokenMixer(X):
    # patch-level (Euclidean)
    local_patches = windows_partition(X)
    patch_centers = avg_pool(local_patches)
    euclidean_dist = cosine_sim(local_patches, patch_centers)
    euclidean_assignment = soft_assign(euclidean_dist, local_proposals)
    patch_feats = token_mixing(local_patches, patch_centers, euclidean_assignment)

    # window-level (Hyperbolic)
    windows = window_pool(X)
    window_centers = avg_pool(windows)
    hwindows = expmap(windows)
    hwindow_centers = expmap(window_centers)
    hyperbolic_dist = lorentz_dist(hwindows, hwindow_centers)
    hyperbolic_assignment = soft_assign(hyperbolic_dist, window_centers)
    window_feats = token_mixing(windows, window_centers, hyperbolic_assignment)
    window_feats = upsampling(window_feats)

    X = X + fc(cat([patch_feats, window_feats], dim=-1))

    return out
```

---

## B    MORE IMPLEMENTATION DETAILS

To elaborate, we provide more detailed descriptions about Center Estimation of the *TokenMixer*. Given the input feature embeddings $\mathbf{F} \in \mathbb{R}^{H \times W \times C}$, we first initialize the data points $\mathbf{X} \in \mathbb{R}^{N \times C}$, where $C = H \times W$. The cluster centers $\mathbf{C} \in \mathbb{R}^{M \times C}$ are estimated via a pooling operator, defined as:

$$\mathbf{C}_i = \frac{1}{K^2} \sum_{p=1}^{K} \sum_{q=1}^{K} \mathbf{F}_{[:,r \cdot K + p, c \cdot K + q]}, \quad s.t. \ \ r = \lfloor i/M_w \rfloor, \quad c = i \bmod M_w, \tag{11}$$

where $K$ is the pooling size, $M_w$ denotes the number of cluster centers in a row, *i.e.*, $M_w = W/K$.

## C    MORE DIAGNOSTIC EXPERIMENTS

We conduct additional ablation studies on the curvature in hyperbolic space and the cluster number in window-level clustering mixing. As shown in Table 5a, it can be seen that the variation in curvature exhibits minimal impact. We hypothesize that this may be attributed to the fact that the current head dimension (24 for Tiny, 32 for Small and Large) is already sufficient to capture the relational

structure within the embedding space across varying curvature regimes. As noted in [36], lower dimensions may exhibit greater sensitivity to curvature, as the combination of low dimensionality and high curvature excessively constricts the capacity of the embedding space. Table 5b presents that a higher cluster count in the hyperbolic space yields performance gains, peaking at 16 clusters. This may be because 25 cluster centers are excessive for the number of windows (set to 49 in this paper).

Table 5: More ablative experiments on ImageNet [66] `val`.

| curvature $\kappa$ | top-1(%)↑ | top-5(%)↑ |
|---|---|---|
| 0.1 | 74.9 | 92.4 |
| 1.0 | 75.0 | 92.5 |
| 10.0 | 74.8 | 92.4 |

| cluster number | top-1(%)↑ | top-5(%)↑ |
|---|---|---|
| 9 | 74.9 | 92.4 |
| 16 | 75.0 | 92.5 |
| 25 | 74.6 | 92.3 |

(a) Curvature $\kappa$ in hyperbolic space.  (b) Cluster number in window-level clustering mixing.

Table 6: Efficiency Analysis for Semantic Segmentation of different backbones with Semantic FPN [14] on ADE20K [12] `val`. Throughput (images/s) measured on a single V100 GPU at batch size 8, averaged over last 500 iterations.

| Backbone | | Param. (M) | GFLOPs (G) | Throughputs (images/s) | mIoU(%) ↑ |
|---|---|---|---|---|---|
| CoC-Small/4 | [7][ICLR23] | 17.6M | 33.5 | 115.9 | 36.6 |
| CoC-Medium/4 | [7][ICLR23] | 25.2M | 48.6 | 60.5 | 40.2 |
| FEC-Small | [8][CVPR24] | 9.1M | 25.3 | 54.4 | 35.3 |
| FEC-Base | [8][CVPR24] | 18.0M | 34.3 | 54.2 | 37.7 |
| FEC-Large | [8][CVPR24] | 31.9M | 49.5 | 24.7 | 40.5 |
| HCFormer-Nano | (ours) | 8.8M | 24.0 | 120.5 | 36.8 |
| HCFormer-Tiny | (ours) | 10.3M | 24.8 | 100.9 | 37.3 |
| HCFormer-Small | (ours) | 18.9M | 34.5 | 74.3 | 40.4 |
| HCFormer-Medium | (ours) | 35.7M | 51.7 | 31.7 | 43.31 |

Table 7: Efficiency Analysis for Object detection and instance segmentation using Mask-RCNN [15] on COCO [13] `val2017`. $AP^{box}$ and $AP^{mask}$ denote bounding box AP and mask AP. Throughput (images/s) measured on a single V100 GPU at batch size 8, averaged over last 500 iterations.

| Method | | Param. | GFLOPs (G) | Throughputs (images/s) | $AP^{box}$ | $AP^{mask}$ |
|---|---|---|---|---|---|---|
| CoC-Small/4 | [7][ICLR23] | 33.6M | 212.2 | 50.8 | 35.9 | 33.8 |
| CoC-Small/25 | [7][ICLR23] | 33.6M | 212.2 | 35.3 | 37.5 | 35.4 |
| CoC-Small/49 | [7][ICLR23] | 33.6M | 212.2 | 25.6 | 37.2 | 34.9 |
| CoC-Medium/4 | [7][ICLR23] | 42.1M | 274.3 | 25.2 | 38.6 | 36.1 |
| CoC-Medium/25 | [7][ICLR23] | 42.1M | 274.3 | 15.2 | 40.1 | 37.4 |
| CoC-Medium/49 | [7][ICLR23] | 42.1M | 274.3 | 10.2 | 40.6 | 37.6 |
| FEC-Small | [8][CVPR24] | 24.3M | 178.8 | 31.2 | 35.6 | 33.6 |
| FEC-Base | [8][CVPR24] | 33.1M | 215.8 | 28.3 | 37.9 | 35.5 |
| FEC-Large | [8][CVPR24] | 47.1M | 277.9 | 13.6 | 39.9 | 37.3 |
| HCFormer-Nano | (ours) | 24.0M | 173.5 | 54.9 | 36.0 | 34.0 |
| HCFormer-Tiny | (ours) | 25.5M | 176.7 | 46.0 | 36.4 | 34.5 |
| HCFormer-Small | (ours) | 34.1M | 216.0 | 33.0 | 38.7 | 36.0 |
| HCFormer-Medium | (ours) | 50.8M | 285.9 | 14.0 | 40.9 | 37.6 |

## D   MORE EFFICIENCY ANALYSIS

As shown in Table 6, our method achieves higher efficiency than FEC and CoC under comparable performance. For instance, HCFORMER-Nano yields 4.6 higher throughput and 11 GFLOPs lower computational cost than CoC-Small/4, while HCFORMER-Small provides 49.6 higher throughput and 15 GFLOPs lower computational cost than FEC-Large. This efficiency advantage becomes more pronounced in object detection/instance segmentation tasks. As indicated in Table 7, the number of cluster centers significantly affects the object detection performance of CoC. When the number increases to 49, CoC exhibits notably lower efficiency than HCFormer. For instance, HCFORMER-Small outperforms CoC-Small/49 by 1.5 and 1.1 in $AP^{box}$ and $AP^{mask}$, respectively, while also

achieving 7.4 higher throughput. Moreover, since FEC employs more cluster centers in downstream tasks compared to its variant for classification, *i.e.*, increasing from [16, 16, 4, 4] to [100, 100, 25, 25], it clearly demonstrates lower efficiency than HCFORMER across all model variants.

# E  MODEL CONFIGURATIONS

We provide the detailed configurations of HCFORMER in Table 8. The settings are aligned with our implementation: Tokenization and inter-stage Downsample are realized by a Point Reducer (Conv2d patch embedding), where patch_size equals stride for the first embedding (4) and uses stride 2 for later downsamples. In ClusterMixer, regions correspond to non-overlapping spatial partitions with regions = fold_w × fold_h, and local_centers correspond to the number of proposal anchors per region with local_centers = proposal_w × proposal_h. In our implementation, the global path is exactly where hyperbolic clustering is applied, while local windows use Euclidean similarity.

Table 8: Model configurations for our Context Cluster. We initialize three variants, *i.e.*, HCFORMER-Tiny, HCFORMER-Small, and HCFORMER-Medium.

| Stage | Input Size | Block | HCFormer-Tiny | | HCFormer-Small | | HCFormer-Medium | |
|---|---|---|---|---|---|---|---|---|
| S1 | 224×224 | Tokenization | $patch\_size = 4$ $stride = 4$ $dim = 32$ | | $patch\_size = 4$ $stride = 4$ $dim = 64$ | | $patch\_size = 4$ $stride = 4$ $dim = 64$ | |
| | 56×56 | ClusterMixer | $regions = 49$ $local\_centers = 16$ $global\_centers = 16$ $heads = 4$ $head\_dim = 24$ $global = \text{True}$ $mlp\_r. = 8$ $dim = 32$ | ×2 | $regions = 49$ $local\_centers = 16$ $global\_centers = 16$ $heads = 4$ $head\_dim = 32$ $global = \text{True}$ $mlp\_r. = 8$ $dim = 64$ | ×2 | $regions = 49$ $local\_centers = 16$ $global\_centers = 16$ $heads = 6$ $head\_dim = 32$ $global = \text{True}$ $mlp\_r. = 8$ $dim = 64$ | ×4 |
| S2 | 56×56 | Downsample | $patch\_size = 3$ $stride = 2$ $padding = 1$ $dim = 64$ | | $patch\_size = 3$ $stride = 2$ $padding = 1$ $dim = 128$ | | $patch\_size = 3$ $stride = 2$ $padding = 1$ $dim = 128$ | |
| | 28×28 | ClusterMixer | $regions = 49$ $local\_centers = 4$ $global\_centers = 16$ $heads = 4$ $head\_dim = 24$ $global = \text{True}$ $mlp\_r. = 8$ $dim = 64$ | ×2 | $regions = 49$ $local\_centers = 4$ $global\_centers = 16$ $heads = 4$ $head\_dim = 32$ $global = \text{True}$ $mlp\_r. = 8$ $dim = 128$ | ×2 | $regions = 49$ $local\_centers = 4$ $global\_centers = 16$ $heads = 6$ $head\_dim = 32$ $global = \text{True}$ $mlp\_r. = 8$ $dim = 128$ | ×4 |
| S3 | 28×28 | Downsample | $patch\_size = 3$ $stride = 2$ $padding = 1$ $dim = 196$ | | $patch\_size = 3$ $stride = 2$ $padding = 1$ $dim = 320$ | | $patch\_size = 3$ $stride = 2$ $padding = 1$ $dim = 320$ | |
| | 14×14 | ClusterMixer | $regions = 1$ $local\_centers = 49$ $heads = 8$ $head\_dim = 24$ $global = \text{False}$ $mlp\_r. = 4$ $dim = 196$ | ×6 | $regions = 1$ $local\_centers = 49$ $heads = 8$ $head\_dim = 32$ $global = \text{False}$ $mlp\_r. = 4$ $dim = 320$ | ×6 | $regions = 1$ $local\_centers = 49$ $heads = 12$ $head\_dim = 32$ $global = \text{False}$ $mlp\_r. = 4$ $dim = 320$ | ×12 |
| S4 | 14×14 | Downsample | $patch\_size = 3$ $stride = 2$ $padding = 1$ $dim = 320$ | | $patch\_size = 3$ $stride = 2$ $padding = 1$ $dim = 512$ | | $patch\_size = 3$ $stride = 2$ $padding = 1$ $dim = 512$ | |
| | 7×7 | ClusterMixer | $regions = 1$ $local\_centers = 16$ $heads = 8$ $head\_dim = 24$ $global = \text{False}$ $mlp\_r. = 4$ $dim = 320$ | ×2 | $regions = 1$ $local\_centers = 16$ $heads = 8$ $head\_dim = 32$ $global = \text{False}$ $mlp\_r. = 4$ $dim = 512$ | ×2 | $regions = 1$ $local\_centers = 16$ $heads = 12$ $head\_dim = 32$ $global = \text{False}$ $mlp\_r. = 4$ $dim = 512$ | ×4 |

# F  DISCUSSION

**Limitation.** One limitation is that the adaptation of our *HCFormer* pre-trained on large-scale datasets (*e.g.*, ImageNet) to downstream tasks (*e.g.*, Semantic Segmentation) may incur accuracy degradation due to domain gaps. For instance, variations in input resolution alter the number of patches per window, which may introduce instability to clustering algorithms responsible for feature aggregation.

Another limitation is that *HCFormer* struggles in handling out-of-distribution or noisy data, which may disrupt the center estimation and cluster assignment. We will dedicate increased efforts to enhancing the in-the-wild robustness of our method in the future direction. Besides, this paper does not conduct comprehensive investigations into hyperparameter configurations such as cluster numbers or window sizes. The model's performance could potentially be further optimized through systematic exploration of these hyperparameters, which warrant further investigation in future work.

**Social Impact.** This work proposes an interpretable and transparent backbone for visual representation learning based on clustering algorithms. On the positive side, this approach is valuable for downstream applications that require understanding of the reasoning process, such as protein design [67] or chemical synthesis [68]. On the negative side, our model may inherit biases in training data through clustering aggregation, raising fairness concerns in sensitive domains such as medical diagnosis or autonomous systems. To mitigate this issue, we recommend implementing a strict security protocol for our method to prevent potential system failures in real-world applications.

**Future work.** In the future, we will explore more powerful, time-efficient clustering techniques to enhance the model's scalability. The incorporation of metric learning to the formulated clusters also holds significant potential for future investigation. Furthermore, we will investigate the application of our HCFormer in downstream tasks involving more complex relational structures, such as scene graph generation and human-object interaction.

# G    LLM USAGE STATEMENT

We employed large language models (LLMs) as auxiliary tools during manuscript preparation. Their use was strictly limited to surface-level editing tasks, including grammar correction, minor rephrasing and stylistic improvements to enhance readability. At no point did we rely on LLMs for generating research ideas, methods, experiments, or conclusions. All technical content and analysis presented in this paper are the sole work of the authors.

