# OpenReview forum: "Hyperbolic Hierarchical Clustering for Visual Representation Learning"
_ICLR.cc/2026/Conference — Submitted to ICLR 2026_

### Official Review · Reviewer_rvV3 · 2025-10-24

**Soundness:** 3
**Presentation:** 3
**Contribution:** 3
**Rating:** 8
**Confidence:** 4

**Summary:**

This paper builds on prior work that uses clustering for token mixing in visual tasks.  The main innovations are to use a hierarchical structure in clustering and to use a hyperbolic space for clustering at a coarser stage.  The work is supported by experiments in visual domains, identification, semantic segmentation and object detection and instance segmentation.  Ablations are included to show the value of these innovations.

**Strengths:**

The paper makes interesting and well motivated modifications to clustering based token mixing.  These seem to lead to real improvements in performance.  The clustering based approach is simpler and intuitive, compared to attention based methods.

**Weaknesses:**

No significant weaknesses, but there are a number of ways in which the clarity of the paper could be improved.
•	Eq. 2 is confusing because you sum a_{i,j} from 1 to M and from 1 to N.  I think something is wrong here.
•	In Eq. 3, what does Norm do?  I at first thought it took the norm of g_i, but that doesn’t seem to make much sense.  Does it normalize it?
•	Some small errors in grammar (eg., We first partitions)
•	Cluster Assignment, might mention clustering approaches that use soft assignment, eg., E-M.
•	This description of Center Estimation could be a little more detailed, maybe in supplementary
•	Descriptions of the results are slightly misleading because the new method is compared to published work that uses somewhat fewer parameters.  It does seem though that the proposed method improves performance.  I know that the number of parameters can’t be equalized, but in some cases the authors could point out that their method achieves the same performance as others with fewer parameters.
•	In ablations, it is confusing because comparisons seem to be to the full model.  It seems like shifted windows should be compared to a hierarchical method with Euclidean distances?  I’m assuming that the Euclidean method that is compared to the hyperbolic one is still hierarchical?
I guess the other limitation of the paper is that the method is not completely original, building closely on prior work, and that the performance increase is somewhat small.  But I still find the results very interesting.

**Questions:**

One of the primary differences between cluster-based token mixing and transformers is that in the cluster-based method the embeddings are compared directly, whereas in transformers they are compared through a query and key, and also combined through a value matrix.  It would be interesting to have some discussion of this issue.  Does their work imply that values, queries and keys are not necessary?  Is there literature on this issue?  Could there be ablations that show that dropping the value, query and key does not hurt performance?  Or is there some reason that they are needed in transformers but not in clustering?  I don’t really expect the authors to address this in the rebuttal, but I find it curious.

---

> ### Author Response · Authors · 2025-11-25
>
> We thank reviewer rvV3 for the valuable time and constructive feedback. We provide point-to-point response below.
>
> ----
>
> ## Reviewer: **rvV3**
>
> **Q1: Eq. 2 is confusing because you sum a_{i,j} from 1 to M and from 1 to N.**
>
> **A1**: We apologize for this mistake and have corrected Eq. 2 in the revision as follows:
>
> $$
> g_i = \sum_{i=1}^{M} \textbf{A} _{i,j} \cdot c_i^{\prime},~~~~\textit{s.t.}~~c_j^{\prime} = (c_j + \sum{i=1}^{N} \textbf{A}{i,j} \cdot x_i)/(1+\sum{i=1}^{N} \textbf{A} _{i,j}).
> $$
>
> **Q2: In Eq. 3, what does Norm do? I at first thought it took the norm of g_i, but that doesn’t seem to make much sense. Does it normalize it?**
>
> **A2**: We thank the reviewer for pointing this out. The term "norm" (referring to the normalization operation) is a typo, which has been removed in the revision of Eq. 3 as follows:
>
> $$
> x_i^{\prime} = x_i + \text{FC}(g_i).
> $$
>
> **Q3: Some small errors in grammar (eg., We first partitions).**
>
> **A3**: Thanks for your suggestion. We have thoroughly reviewed the entire manuscript and corrected the grammatical errors highlighted in the revision.
>
>
> **Q4: Cluster Assignment, might mention clustering approaches that use soft assignment, eg., E-M.**
>
> **A4**: Thank you for raising this point. We have augmented the revision with the following content:
>
> > L192-195: This process resembles the the Expectation-step (E-step) in Expectation-Maximization (EM) clustering [51], with the distinction that our cluster centers are precomputed via average pooling over neighboring patches, avoiding iterative updates.
>
>
> **Q5: This description of Center Estimation could be a little more detailed, maybe in supplementary.**
>
> **A5**: We have added the more details about center estimation to the supplementary materials in the revised manuscript, as follows:
>
> > L742-749: To elaborate, we provide more detailed descriptions about Center Estimation of the \textit{TokenMixer}.
> Given the input feature embeddings $\textbf{F}\!\in\!\mathbb{R}^{H \times W \times C}$, we first initialize the data points $\textbf{X}\!\in\!\mathbb{R}^{N \times C}$, where $C\!=\!H \times W$. The cluster centers $\textbf{C}\!\in\!\mathbb{R}^{M \times C} $ is estimated via a pooling operator, defined as:
>
> $$
> \textbf{C} _{i} = \frac{1}{K^2}\sum^{K} _{p=1}\sum^{K} _{q=1} \textbf{F} _{[:, r \cdot K + p, c \cdot K + q]},~~~~\textit{s.t.}~~r = \left\lfloor i/M_w \right\rfloor, \quad c = i \bmod M_w,
> $$
>
> > L742-749: where $K$ is the pooling size, $M_w$ denotes the number of cluster centers in a row, \ie, $M_w\!=\!W/K$.
>
> **Q6: Descriptions of the results are slightly misleading because the new method is compared to published work that uses somewhat fewer parameters.  It does seem though that the proposed method improves performance.  I know that the number of parameters can’t be equalized, but in some cases the authors could point out that their method achieves the same performance as others with fewer parameters.**
>
> **A6**: Apologies for the confusion. To further demonstrate the model's performance, we conducted additional experiments with a smaller-parameter variant, as summarized in the revised table below:
>
> | Method          | Param. (M) | GFLOPs (G) | Top-1 | Throughputs (images/s) |
> | :-------------- | :--------- | :--------- | :---- | :--------------------- |
> | HCFormer-Nano   | 5.1  | 0.9 | 73.1 | 719.0 |
> | HCFormer-Tiny   | 7.0  | 1.0 | 75.0 | 536.0 |
> | HCFormer-Small  | 16.1 | 2.9 | 79.9 | 324.7 |
> | HCFormer-Medium | 33.7 | 6.4 | 82.4 | 235.7 |
>
> Additionally, in response to the reviewer's suggestions, we have revised the Results sections accordingly.
>
> > L322-355: To further substantiate the performance of our method, we introduce a smaller variant of HCFormer with only 5.1M parameters, i.e., HCFormer-Nano, which has fewer parameters than all other methods listed in Table 1. Despite its compact size, HCFormer-Nano achieves 73.1% top-1 accuracy, exceeding CoC-Tiny by 1.3% with 0.2M fewer parameters and FEC-Small by 0.4% with 0.4M fewer parameters.
>
> > L398-401: Notably, HCFormer-Small achieves performance comparable to CoC-Medium/4 and FEC-Large, with 6.3M and 13.0M fewer parameters, respectively. Furthermore, our HCFormer-Nano outperforms FEC-Small by 1.5% with 0.3M fewer parameters, while even surpassing CoC-Small (17.6M parameters).
>
> > L422-423: With fewer parameters, HCFormer-Nano outperforms FEC-Small by 0.4% $\text{AP}^\text{box}$ and 0.4% $\text{AP}^\text{mask}$.

---

> ### Author Response · Authors · 2025-11-25
>
> **Q7: In ablations, it is confusing because comparisons seem to be to the full model. It seems like shifted windows should be compared to a hierarchical method with Euclidean distances? I’m assuming that the Euclidean method that is compared to the hyperbolic one is still hierarchical?**
>
> **A7**: Our apologies. Accordingly, we have appropriately revised Table 4a and the corresponding analysis of the results in the revised manuscript.
>
> | Strtegy                 | top-1(%)$\uparrow$ | top-5(%)$\uparrow$ |
> | :-----------------------| :----------------- | :----------------- |
> | Shifted Windows         | 72.7%              | 91.1%              |
> | Hierarchical Clustering | 73.4%              | 91.4%              |
>
> > L472-474:  As outlined in Table 1a, while the shifted window mechanism remains effective, our hierarchical clustering strategies are better suited for \textit{ClusterMixer} (\ie, 72.7\% $\rightarrow$ 73.4\%)
>
>
> **Q8: I guess the other limitation of the paper is that the method is not completely original, building closely on prior work, and that the performance increase is somewhat small. But I still find the results very interesting.**
>
> **A8**: We thank the reviewer for the positive feedback.
> Our work is indeed inspired by CoC [ref1], which introduces clustering algorithms into visual representation learning.
> Given its interpretable nature, we believe clustering-based vision backbones represent a promising research direction.
> However, their current limited performance hinders broader application.
> Therefore, we aim to further explore the potential of clustering algorithms to develop clustering-based vision backbones that achieve a more favorable balance between interpretability and performance.
> Our goal is to attract attention from the vision community and contribute to a paradigm shift toward such methods.
>
> [ref1] Image as Set of Points. ICLR23.
>
>
> **Q9: One of the primary differences between cluster-based token mixing and transformers is that in the cluster-based method the embeddings are compared directly, whereas in transformers they are compared through a query and key, and also combined through a value matrix. It would be interesting to have some discussion of this issue. Does their work imply that values, queries and keys are not necessary? Is there literature on this issue? Could there be ablations that show that dropping the value, query and key does not hurt performance? Or is there some reason that they are needed in transformers but not in clustering? I don’t really expect the authors to address this in the rebuttal, but I find it curious.**
>
> **A9**: We thank the reviewer for this insightful comment.
> A growing body of work indicates that self-attention (QKV) is not essential for certain tasks.
> For instance, AFNO [ref2] replaces self-attention with adaptive token mixing in the Fourier domain, achieving superior efficiency and performance in semantic segmentation.
> Similarly, Yu et al. [ref3] show that a MetaFormer using only separable convolutions (ConvFormer) outperforms Swin Transformer on ImageNet-1K.
> PointMixer [ref4] substitutes complex self-attention with simple MLP and Softmax operations, achieving comparable or better performance with greater parameter efficiency, faster training, and improved generalization in point cloud processing.
> Furthermore, [ref5] demonstrates that in specialized domains like quantum wave function modeling, the query and key components are largely superfluous, with their removal resulting in negligible accuracy loss.
> These findings suggest that the QKV mechanism is not essential for achieving high performance on tasks where the modeling priority lies in capturing local feature interactions or structural patterns rather than long-range, global context.
> In such contexts, it can be effectively supplanted by clustering algorithms or even simpler token mixing operations like average pooling.
> However, due to its flexibility and expressive power, QKV mechanisms remain essential in transformers for tasks requiring complex, context-aware reasoning (e.g., machine translation and complex question answering) rather than clustering.
>
>
> [ref2] Adaptive Fourier Neural Operators: Efficient Token Mixers for Transformers. ICLR21.
>
> [ref3] MetaFormer Baselines for Vision. TPAMI23.
>
> [ref4] PointMixer: MLP-Mixer for Point Cloud Understanding.
>
> [ref5] Are queries and keys always relevant? A case study on transformer wave functions. Machine Learning: Science and Technology, 2025.

---

> > ### Comment · Reviewer_rvV3 · 2025-11-26
> > **Thanks**
> >
> > I thank the authors for their comprehensive responses.

---

### Official Review · Reviewer_UABQ · 2025-10-27

**Soundness:** 3
**Presentation:** 4
**Contribution:** 3
**Rating:** 6
**Confidence:** 4

**Summary:**

This paper proposes HCFormer (Hyperbolic Hierarchical Clustering Transformer), a novel vision backbone that replaces self-attention with a hierarchical clustering-based token mixer. The key idea is to first perform local Euclidean clustering at the patch level, followed by global hyperbolic clustering across window-level features. The authors argue that hyperbolic geometry (negative curvature) naturally captures hierarchical semantics, offering both representational compactness and computational efficiency. Experimental results on ImageNet, COCO, and ADE20K show consistent improvements over CoC and FEC under comparable parameter and FLOP budgets.

**Strengths:**

- **Conceptual novelty**: Introducing hyperbolic geometry into a hierarchical clustering-based vision transformer is a fresh and well-motivated idea. The connection between tree-like semantic structures and negative curvature is theoretically sound and intuitively clear.

- **Intuitive architecture**: The design cleanly unifies local Euclidean and global hyperbolic clustering. The structure is modular and can be plugged into standard ViT-style backbones.

- **Strong experimental coverage**: Evaluations span classification, detection, and segmentation. Gains of +0.7–1.0% Top-1 on ImageNet and consistent improvements on COCO/ADE20K demonstrate robustness.

- **Comprehensive ablations**: The paper includes clear ablations showing the independent and combined effects of hierarchical structure (+0.8–1.0%) and hyperbolic space (+0.5–0.7%).

- **Presentation quality**: The paper is very well written, with clear figures and consistent notation. Methodology and empirical sections are easy to follow.

**Weaknesses:**

* **Efficiency claim not fully supported.**
  While the paper theoretically reduces complexity from ($O(N^2)$) (self-attention) to ($O(NM)$) via hierarchical clustering, the *measured* FLOPs and throughput remain almost identical to CoC and FEC. The hyperbolic distance computation (arcosh + mapping functions) introduces overhead that cancels out the theoretical benefit.

* **Performance gains are modest.**
The reported gains are steady but incremental. In Table 1, HCFormer outperforms the prior FEC baseline by roughly +0.7–1.0 Top-1 % on ImageNet-1K under similar FLOPs and parameter budgets. The largest relative gap (Tiny → FEC-Small) is +2.3 points, while improvements shrink at larger scales (+1.2 points for Medium). Ablations (Table 4b–c) show that hierarchical clustering contributes about +2.3 points, whereas the hyperbolic geometry adds +1.6 points in a 6 M-parameter setting. However, the hyperbolic benefit is not shown for larger backbones or downstream tasks, and no analysis is provided on curvature sensitivity or numerical stability. Hence, while the geometric component is empirically beneficial, its effect remains secondary and under-validated

* **Limited analysis of hyperbolic embedding.**
  There is no exploration of curvature sensitivity, embedding radius, or gradient stability of the exp/log mappings. Without such analysis, the hyperbolic advantage remains somewhat qualitative.

* **Sec. 3.2 Cluster Mixer Writing Clarification**.
Section 3.2 primarily restates the CoC/FEC-style token–cluster mixing operation (soft assignment → aggregation → redistribution). The real novelty appears in Sections 3.3 and 3.4, which introduce hierarchical organization and hyperbolic distance. Not sure the claim of a new token mixer from the authors is legitimate.

**Questions:**

1. How sensitive are the results to the number of clusters (M) and the curvature parameter of the hyperbolic space?
2. Have you measured actual runtime efficiency (throughput) compared to FEC/CoC on identical hardware?
3. Could the same hierarchical clustering structure achieve similar benefits in Euclidean space with proper scaling or depth adjustments?

---

> ### Author Response · Authors · 2025-11-25
>
> We thank reviewer UABQ for the valuable time and constructive feedback. We provide point-to-point response below.
>
> ----
>
> ## Reviewer: **UABQ**
>
> **Q1: Efficiency claim not fully supported. While the paper theoretically reduces complexity from $O(N^2)$ (self-attention) to $O(NM)$ via hierarchical clustering, the measured FLOPs and throughput remain almost identical to CoC and FEC. The hyperbolic distance computation (arcosh + mapping functions) introduces overhead that cancels out the theoretical benefit.**
>
> **A1**: Thanks for your insightful comments. We acknowledge that the throughput of our method is nearly identical to that of CoC and FEC (as shown in Table 1). Nevertheless, when applied to downstream tasks (semantic segmentation, object detection/instance segmentation), our method demonstrates significantly higher efficiency.
>
> The results for semantic segmentation are presented below:
>
> | Method | Param. (M) | GFLOPs (G) | Throughputs (images/s) | mIoU(%)$\uparrow$ |
> | :- | :- | :- | :- | :- |
> | CoC-Small/4  | 17.6 | 33.5 | 115.9 | 36.6 |
> | CoC-Medium/4 | 25.2 | 48.6 | 60.5 | 40.2 |
> | FEC-Small    | 9.1  | 25.3 | 54.4 | 35.3 |
> | FEC-Base     | 18.0 | 34.3 | 54.2 | 37.7 |
> | FEC-Large    | 31.9 | 49.5 | 24.7 | 40.5 |
> | HCFormer-Nano   | 8.8  | 24.0 | 120.5 | 36.8 |
> | HCFormer-Tiny   | 10.3 | 24.8 | 100.9 | 37.3 |
> | HCFormer-Small  | 16.1 | 34.5 | 74.3 | 40.4 |
> | HCFormer-Medium | 33.7 | 51.7 | 31.7 | 43.3 |
>
> As seen, our method achieves higher efficiency than FEC and CoC under comparable performance. For instance, HCFormer-Nano yields 4.6 higher throughput and 11 GFLOPs lower computational cost than CoC-Small/4, while HCFormer-Small provides 49.6 higher throughput and 15 GFLOPs lower computational cost than FEC-Large.
>
> The results for object detection/instance segmentation are presented below:
>
> | Method | Param. (M) | GFLOPs (G) | Throughputs (images/s) | $\text{AP}^\text{box}\uparrow$ | $\text{AP}^\text{mask}\uparrow$ |
> | :- | :- | :- | :- | :- | :- |
> | CoC-Small/4  | 33.6 | 212.2 | 50.8 | 35.9 | 33.8 |
> | CoC-Small/25 | 33.6 | 212.2 | 35.3 | 37.5 | 35.4 |
> | CoC-Small/49 | 33.6 | 212.2 | 25.6 | 37.2 | 34.9 |
> | CoC-Medium/4 | 42.1 | 274.3 | 25.2 | 38.6 | 36.1 |
> | CoC-Medium/25| 42.1 | 274.3 | 15.2 | 40.1 | 37.4 |
> | CoC-Medium/49| 42.1 | 274.3 | 10.2 | 40.6 | 37.6 |
> | FEC-Small    | 24.3 | 178.8 | 31.2 | 35.6 | 33.6 |
> | FEC-Base     | 33.1 | 215.8 | 28.3 | 37.9 | 35.5 |
> | FEC-Large    | 47.1 | 277.9 | 13.6 | 39.9 | 37.3 |
> | HCFormer-Nano   | 24.0 | 173.5 | 54.9 | 36.0 | 34.0 |
> | HCFormer-Tiny   | 25.5 | 176.7 | 46.0 | 36.4 | 34.5 |
> | HCFormer-Small  | 34.1 | 216.0 | 33.0 | 38.7 | 36.0 |
> | HCFormer-Medium | 50.8 | 285.9 | 14.0 | 40.9 | 37.6 |
>
> It can be seen that the number of cluster centers significantly affects the object detection performance of CoC. When the number increases to 49, CoC exhibits notably lower efficiency than HCFormer. For instance, HCFormer-Small outperforms CoC-Small/49 by 1.5 and 1.1 in $\text{AP}^\text{box}$ and $\text{AP}^\text{mask}$, respectively, while also achieving 7.4 higher throughput.Moreover, since FEC employs more cluster centers in downstream tasks compared to its variant for classification, i.e., increasing from [16, 16, 4, 4] to [100, 100, 25, 25], it clearly demonstrates lower efficiency than HCFormer across all model variants.
>
> This discussion has been incorporated into the revision:
>
> > L803-812: As shown in Table 6, our method achieves higher efficiency than FEC and CoC under comparable performance. For instance, HCFormer-Nano yields 4.6 higher throughput and 11 GFLOPs lower computational cost than CoC-Small/4, while HCFormer-Small provides 49.6 higher throughput and 15 GFLOPs lower computational cost than FEC-Large. This efficiency advantage becomes more pronounced in object detection/instance segmentation tasks. As indicated in Table 7, the number of cluster centers significantly affects the object detection performance of CoC. When the number increases to 49, CoC exhibits notably lower efficiency than HCFormer. For instance, HCFormer-Small outperforms CoC-Small/49 by 1.5 and 1.1 in $\text{AP}^\text{box}$ and $\text{AP}^\text{mask}$, respectively, while also achieving 7.4 higher throughput. Moreover, since FEC employs more cluster centers in downstream tasks compared to its variant for classification, i.e., increasing from [16, 16, 4, 4] to [100, 100, 25, 25], it clearly demonstrates lower efficiency than HCFormer across all model variants.
>
> Thanks once again for the reviewer’s feedback. In response, we have revised the efficiency claim in the Introduction as detailed below:
>
> > L86-88: This reduction in complexity is especially beneficial for downstream tasks, e.g., semantic segmentation and object detection, which facilitates the use of a greater number of cluster centers for improved performance

---

> ### Author Response · Authors · 2025-11-25
>
> **Q2: Performance gains are modest. The reported gains are steady but incremental. In Table 1, HCFormer outperforms the prior FEC baseline by roughly +0.7–1.0 Top-1 % on ImageNet-1K under similar FLOPs and parameter budgets. The largest relative gap (Tiny → FEC-Small) is +2.3 points, while improvements shrink at larger scales (+1.2 points for Medium).**
>
> **A2**: Thank the reviewer for this insightful comment.
> HCFormer demonstrates a steady and meaningful performance advantage over FEC on ImageNet (2.3%/1.8%/1.2%), which, although slightly narrowing at larger scales, remains considerable on the challenging ImageNet-1k benchmark.
> Moreover，our method achieves **substantial improvements on downstream tasks**, and **these gains do not diminish at larger scales**. For example, HCFormer-Small outperforms FEC-base by 2.7% mIoU, while HCFormer-Medium still exceeds FEC-large by 2.8% mIoU. In object detection, HCFormer-Small surpasses FEC-base by 0.8% $\text{AP}^{box}$, and HCFormer-Medium maintains a lead over FEC-large by 1.0% $\text{AP}^{box}$. Notably, it is promising that our model shows the potential to achieve performance on par with or even exceeding that of the Swin Transformer under similar parameter budgets, as illustrated by the fact that our performance curve lies above Swin Transformer's in Figure 1. **This outcome holds meaningful implications for cluster-based architectures, suggesting that the performance limitations often associated with such approaches may have been partially alleviated.**
>
>
> **Q3: Ablations (Table 4b–c) show that hierarchical clustering contributes about +2.3 points, whereas the hyperbolic geometry adds +1.6 points in a 6 M-parameter setting. However, the hyperbolic benefit is not shown for larger backbones or downstream tasks, and no analysis is provided on curvature sensitivity or numerical stability. Hence, while the geometric component is empirically beneficial, its effect remains secondary and under-validated.**
>
> **A3**: To address your concerns, we have included the results of ablation studies on downstream tasks:
>
> | Hier. Clus. | Hyp. Geo. | Rel. Pos. | top-1(%)$\uparrow$ | top-5(%)$\uparrow$ | mIoU(%)$\uparrow$ | $\text{AP}^{box}\uparrow$ | $\text{AP}^{mask}\uparrow$ |
> | :---------- | :-------- | :-------- | :----------------- | :----------------- | :---------------- | :---- | :---- |
> |   |   |   | 71.9 | 90.8 | 35.1 | 34.3 | 32.7 |
> |   | ✓ | ✓ | 72.7 | 91.1 | 34.6 | 34.4 | 32.8 |
> | ✓ |   | ✓ | 73.4 | 91.4 | 35.2 | 34.7 | 33.1 |
> | ✓ | ✓ |   | 74.4 | 92.2 | 36.4 | 35.9 | 34.2 |
> | ✓ | ✓ | ✓ | 75.0 | 92.5 | 37.3 | 36.4 | 34.5 |
>
> As observed, the hyperbolic benefit is also evident in downstream tasks.
>
> Furthermore, we also have included experiments to investigate curvature sensitivity, as:
>
> | curvature  | top-1(%)$\uparrow$ | top-5(%)$\uparrow$ |
> | :--------- | :---- | :---- |
> | K=0.1      | 74.9  | 92.4  |
> | K=1.0      | 75.0  | 92.5  |
> | K=10.0     | 74.8  | 92.4  |
>
> It can be seen that the variation in curvature exhibits minimal impact. We hypothesize that this may be attributed to the fact that the current head dimension (24 for Tiny, 32 for Small and Large) is already sufficient to capture the relational structure within the embedding space across varying curvature regimes. As noted in [ref1], lower dimensions may exhibit greater sensitivity to curvature, as the combination of low dimensionality and high curvature excessively constricts the capacity of the embedding space.
>
> This discussion has been incorporated into the revision:
>
> > L754-758: As shown in Table 5a, it can be seen that the variation in curvature exhibits minimal impact. We hypothesize that this may be attributed to the fact that the current head dimension (24 for Tiny, 32 for Small and Large) is already sufficient to capture the relational structure within the embedding space across varying curvature regimes. As noted in [36], lower dimensions may exhibit greater sensitivity to curvature, as the combination of low dimensionality and high curvature excessively constricts the capacity of the embedding space.
>
> I'm sorry, but due to hardware limitations, we are unable to provide ablation results for larger configurations of HCFormer during the rebuttal period.
>
> [ref1]: Hyperbolic Image Segmentation. CVPR22.

---

> ### Author Response · Authors · 2025-11-25
>
> **Q4: Limited analysis of hyperbolic embedding. There is no exploration of curvature sensitivity, embedding radius, or gradient stability of the exp/log mappings. Without such analysis, the hyperbolic advantage remains somewhat qualitative.**
>
> **A4**: In response to your concern, we present experimental results under varying curvature settings, as:
>
> | curvature  | top-1(%)$\uparrow$ | top-5(%)$\uparrow$ |
> | :--------- | :---- | :---- |
> | K=0.1      | 74.9  | 92.4  |
> | K=1.0      | 75.0  | 92.5  |
> | K=10.0     | 74.8  | 92.4  |
>
> It can be seen that the variation in curvature exhibits minimal impact. We hypothesize that this may be attributed to the fact that the current head dimension (24 for Tiny, 32 for Small and Large) is already sufficient to capture the relational structure within the embedding space across varying curvature regimes. As noted in [ref1], lower dimensions may exhibit greater sensitivity to curvature, as the combination of low dimensionality and high curvature excessively constricts the capacity of the embedding space.
>
> This discussion has been incorporated into the revision:
>
> > L754-758: As shown in Table 5a, it can be seen that the variation in curvature exhibits minimal impact. We hypothesize that this may be attributed to the fact that the current head dimension (24 for Tiny, 32 for Small and Large) is already sufficient to capture the relational structure within the embedding space across varying curvature regimes. As noted in [36], lower dimensions may exhibit greater sensitivity to curvature, as the combination of low dimensionality and high curvature excessively constricts the capacity of the embedding space.
>
> [ref1]: Hyperbolic Image Segmentation. CVPR22.
>
>
> **Q5: Sec. 3.2 Cluster Mixer Writing Clarification. Section 3.2 primarily restates the CoC/FEC-style token–cluster mixing operation (soft assignment → aggregation → redistribution). The real novelty appears in Sections 3.3 and 3.4, which introduce hierarchical organization and hyperbolic distance. Not sure the claim of a new token mixer from the authors is legitimate.**
>
> **A5**: Thank you for your comment and for highlighting the contributions of Sections 3.3 and 3.4. We agree that the clustering workflow in Section 3.2 resembles those in CoC and FEC. We wish to clarify that our core contribution in this section is the conceptual reformulation of the clustering operation itself as a standalone, general-purpose token mixer. While the computational steps may appear similar, this reformulation is implemented with two key distinctions that enhance its capability as a token-mixing module (as stated in L208-211):
> 1. the soft clustering (v.s. CoC's hard clustering) enables our method to benefit from constructing more clusters across various tasks.
> 2. the relative positional bias effectively assists the clustering algorithm in establishing relational dependencies.
>
> This explicit formalization of the "cluster mixer" is essential for guiding our subsequent strategy design in Sections 3.3 and 3.4, and also establishes a principled paradigm for developing clustering-based vision backbones.
>
> To clarify this point, we have revised the manuscript as follows:
>
> > L18-19: Diverging from these opaque designs, we introduce ClusterMixer, a transparent token mixer that is grounded in a clustering paradigm and interpretable by design.
>
> > L54-55: In this work, we introduce an efficient token mixer based on clustering algorithms, dubbed ClusterMixer, which operates via: ...
>
> We sincerely appreciate your valuable feedback, which has helped us improve the clarity of our manuscript.

---

> ### Author Response · Authors · 2025-11-25
>
> **Q6: How sensitive are the results to the number of clusters (M) and the curvature parameter of the hyperbolic space?**
>
> **A6**: In response to your concern, we present experimental results under varying curvature settings, as:
>
> | curvature  | top-1(%)$\uparrow$ | top-5(%)$\uparrow$ |
> | :--------- | :---- | :---- |
> | K=0.1      | 74.9  | 92.4  |
> | K=1.0      | 75.0  | 92.5  |
> | K=10.0     | 74.8  | 92.4  |
>
> It can be seen that the variation in curvature exhibits minimal impact. We hypothesize that this may be attributed to the fact that the current head dimension (24 for Tiny, 32 for Small and Large) is already sufficient to capture the relational structure within the embedding space across varying curvature regimes. As noted in [ref1], lower dimensions may exhibit greater sensitivity to curvature, as the combination of low dimensionality and high curvature excessively constricts the capacity of the embedding space.
>
> Furthermore, we present the results of varying the number of clusters (M) in the hyperbolic space, as:
>
> | cluster number  | top-1(%)$\uparrow$ | top-5(%)$\uparrow$ |
> | :-------------- | :---- | :---- |
> | 9   | 74.9 | 92.4 |
> | 16  | 75.0 | 92.5 |
> | 25  | 74.6 | 92.3 |
>
> It can be seen that a higher cluster count in the hyperbolic space yields performance gains, peaking at 16 clusters. This may be because 25 cluster centers are excessive for the number of windows (set to 49 in this paper).
>
> This discussion has been incorporated into the revision:
>
> > L754-758: As shown in Table 5a, it can be seen that the variation in curvature exhibits minimal impact. We hypothesize that this may be attributed to the fact that the current head dimension (24 for Tiny, 32 for Small and Large) is already sufficient to capture the relational structure within the embedding space across varying curvature regimes. As noted in [36], lower dimensions may exhibit greater sensitivity to curvature, as the combination of low dimensionality and high curvature excessively constricts the capacity of the embedding space.
>
> > L758-760: Table 5b presents that a higher cluster count in the hyperbolic space yields performance gains, peaking at 16 clusters. This may be because 25 cluster centers are excessive for the number of windows (set to 49 in this paper).
>
>
> [ref1]: Hyperbolic Image Segmentation. CVPR22.
>
>
> **Q7: Have you measured actual runtime efficiency (throughput) compared to FEC/CoC on identical hardware?**
>
> **A7**: Thank you for pointing this out. We have re-measured the throughput of FEC and CoC on the identical hardware, as follows:
>
> | Method          | Param. (M) | Throughputs (images/s) |
> | :-------------  | :--------- | :--------------------- |
> | CoC-tiny        | 5.3  | 792.5 |
> | CoC-Small       | 14.0 | 581.8 |
> | CoC-Medium      | 27.9 | 473.8 |
> | FEC-Small       | 5.5  | 742.1 |
> | FEC-Base        | 14.4 | 532.1 |
> | FEC-Large       | 28.3 | 478.2 |
> | HCFormer-Nano   | 5.1  | 719.0 |
> | HCFormer-Tiny   | 7.0  | 536.0 |
> | HCFormer-Small  | 16.1 | 324.7 |
> | HCFormer-Medium | 33.7 | 235.7 |
>
> These measurements have been updated in the revision. For fair comparison, the throughputs of other methods have been temporarily removed and will be re-measured in a subsequent version.
>
>
> **Q8: Could the same hierarchical clustering structure achieve similar benefits in Euclidean space with proper scaling or depth adjustments?**
>
> **A8**: That's an interesting point.
> Experimental results (Table 6) from [ref2] indicate that hyperbolic embeddings exhibit a marked performance advantage over Euclidean embeddings in lower dimensions, while this advantage diminishes with increasing dimensionality. This suggests that increasing the dimensionality of Euclidean embeddings may approximate the representational efficiency of hyperbolic space.
> However, [ref3] raises the concern that as semantic complexity increases (such as in richly structured visual scenes), Euclidean embeddings require an arbitrarily large number of dimensions to capture the data faithfully, whereas hyperbolic space maintains high representational capacity with consistently low dimensionality.
> Therefore, systematically investigating in which task scenarios and how to adjust dimensions so that Euclidean space can achieve effects comparable to hyperbolic space warrants further exploration and substantial effort. We are keen to pursue this as a future research direction.
>
> [ref2] Hyperbolic Image Embeddings. CVPR20.
>
> [ref3] Hyperbolic Contrastive Learning for Visual Representations beyond Objects. CVPR23.
>
> [ref4] Hyperbolic Representation Learning: Revisiting and Advancing. ICML23.

---

### Official Review · Reviewer_fjaH · 2025-11-01

**Soundness:** 2
**Presentation:** 1
**Contribution:** 2
**Rating:** 2
**Confidence:** 2

**Summary:**

The paper proposes HCFormer, a MetaFormer-style vision backbone implementing ClusterMixer for token mixing via hierarchical clustering across dual geometries: Euclidean space for patch-level clustering within local windows and hyperbolic (Lorentz) space for window-level global abstraction. While claiming interpretability and efficiency advantages, the paper demonstrates competitive results on ImageNet, ADE20K, and COCO benchmarks but suffers from fundamental technical inconsistencies and unsupported theoretical claims.

**Strengths:**

1. Novel architectural concept: Dual-geometry approach (Euclidean for fine-grained, hyperbolic for hierarchical) is creative
2. Comprehensive evaluation: Experiments across ImageNet, ADE20K, COCO with consistent improvements over clustering baselines
3. Competitive performance, comparable to attention-based models

**Weaknesses:**

Major
1. Fundamental design flaw in dual-geometry integration: in the paper, author concatenates features from Euclidean and hyperbolic spaces directly in Eq. (10) without any alignment mechanism. This is mathematically problematic as these features exist in incompatible metric spaces with different scales and properties. The simple concatenation likely destroys the geometric structure that hyperbolic space is supposed to preserve, undermining the entire theoretical motivation.
 - scale: Euclidean cosine similarity ranges in [-1, 1] while Lorentz distance has unbounded range. Without proper normalization, one component will dominate the Softmax computation.
 - Typically, when combining features from different manifolds, you need to either: a). Project both to a common space (e.g., tangent space)
 b) Use separate processing streams with late fusion, c) Apply learned projection/alignment layers
 - Cosine similarity (higher=better) vs Lorentz distance (lower=better) both fed to Softmax without explaining sign conversion
2. Why are window-level relationships hierarchical? No explanation or evidence provided
3. No evidence showing what hyperbolic geometry actually learns or why it helps
4. Only one visualization provided, no quantitative interpretability metrics (cluster purity, consistency, semantic alignment)
5. The paper completely omits standard practices for hyperbolic neural networks: No dimension reduction before hyperbolic operations (standard practice to reduce from C to C/r for computational efficiency [Khrulkov et al. 2020]:)

Minor
1. Eq. (10) notation error: g^W_L suggests window-Lorentz while g^P_E suggests patch-Euclidean, directly contradicting the text's claim of patch=Euclidean, window=hyperbolic
2. Eq. (2) indexing bug: Uses c'_i but should be c'_j
3. No exploration of cluster numbers, window sizes, or curvature parameter κ.

**Questions:**

1. How is your method actually implementable? Please provide the exact dimensions at each step: C → ? (reduction?) → hyperbolic (C+1?) → ? (projection?) → concatenation → FC. Without these details, the method cannot be reproduced.
2. What is the true parameter count? If FC reduces (2C+1) → C after concatenation, this adds significant parameters. Why isn't this in Table 1?
3. How do you handle high-dimensional hyperbolic operations? Processing 320-512 dims in hyperbolic space without dimension reduction contradicts standard practices. How is this numerically stable?
4. Why not test Euclidean-Euclidean dual-path? This critical control would reveal whether improvements come from hyperbolic geometry or simply dual-path processing.

---

> ### Author Response · Authors · 2025-11-25
>
> We thank reviewer fjaH for the valuable time and constructive feedback. We provide point-to-point response below.
>
> ----
>
> ## Reviewer: fjaH
>
> **Q1: Fundamental design flaw in dual-geometry integration: in the paper, author concatenates features from Euclidean and hyperbolic spaces directly in Eq. (10) without any alignment mechanism.  This is mathematically problematic as these features exist in incompatible metric spaces with different scales and properties. The simple concatenation likely destroys the geometric structure that hyperbolic space is supposed to preserve, undermining the entire theoretical motivation. scale: Euclidean cosine similarity ranges in [-1, 1] while Lorentz distance has unbounded range. Without proper normalization, one component will dominate the Softmax computation. Typically, when combining features from different manifolds, you need to either: a). Project both to a common space (e.g., tangent space) b) Use separate processing streams with late fusion, c) Apply learned projection/alignment layers. Cosine similarity (higher=better) vs Lorentz distance (lower=better) both fed to Softmax without explaining sign conversion**
>
> **A1**: Sorry for this misunderstanding. We would like to clarify that we only define the similarity metric $D$ (Eq. 1) separately in Euclidean space and hyperbolic space.
> In other words, features projected into hyperbolic space are used exclusively for computing the assignment matrix. **There is no direct fusion or combination of features from Euclidean and hyperbolic spaces.** For improved clarity, we have added the following to the revised version:
>
> > L285-286: The computations for both are performed in parallel and remain decoupled from one another.
>
>
> **Q2: Why are window-level relationships hierarchical? No explanation or evidence provided**
>
> **A2**: Sorry for this misunderstanding. As outlined in Section 1 (Lines 69-73), Section 3.3 (Lines 214-215), and Figure 2, our approach is inherently hierarchical by design:  the input image is first partitioned into multiple windows, each of which consists of several patches. Window-level token mixing is performed globally across all windows, while patch-level token mixing is conducted locally within each window. This global-to-local workflow naturally establishes a hierarchical relationship between window-level and patch-level representations.
>
>
> **Q3: No evidence showing what hyperbolic geometry actually learns or why it helps**
>
> **A3**: We thank the reviewer for raising this insightful and critical point.
> In response, we offer both theoretical motivation and empirical evidence to clarify the role and benefits of hyperbolic geometry in our approach.
>
> - **Theoretical motivation**: Unlike Euclidean geometry, which excels in modeling flat and simple structures, hyperbolic geometry naturally captures hierarchical relationships as a continuous analog of trees, thereby more effectively representing abstract and complex semantics.
> In our framework, window-level clustering necessitates the extraction of such abstract semantics.
> Here, hyperbolic geometry maintains its representational capacity with consistently low dimensionality, whereas Euclidean embeddings would require significantly higher dimensions [ref1].
> As semantic complexity grows (such as in richly structured visual scenes), Euclidean embeddings would require an arbitrarily large number of dimensions to faithfully represent the data [ref2].
> Thus, hyperbolic geometry offers a principled and efficient framework for modeling our data.
>
> - **Empirical evidence**: The experimental results presented in Table 4c demonstrate that incorporating hyperbolic geometry yields a significant performance gain of 1.6% in top-1 accuracy compared to its Euclidean counterpart. Furthermore, this improvement is consistently observed across multiple tasks: semantic segmentation (+2.1% mIoU), object detection (+1.7% AP$^{\text{box}}$), and instance segmentation (+1.4% AP$^{\text{mask}}$). These empirical results further validate the benefits of adopting hyperbolic geometry in our approach.
>
> [ref1] Hyperbolic Representation Learning: Revisiting and Advancing. ICML23.
>
> [ref2] Hyperbolic Contrastive Learning for Visual Representations beyond Objects. CVPR23.

---

> ### Author Response · Authors · 2025-11-25
>
> **Q4: Only one visualization provided, no quantitative interpretability metrics (cluster purity, consistency, semantic alignment)**
>
> **A4**: We agree that quantitative metrics are valuable. However, metrics like cluster purity and consistency require ground-truth labels for each individual data point. In our method, each data point corresponds to a 4×4 image patch, and it is currently challenging to identify a suitable dataset that provides dense semantic annotations at this granularity. Furthermore, determining the appropriate number of cluster centers to align with predefined semantic categories requires further investigation. We consider these issues to be beyond the scope of the present work and will address them in future research.
>
> Regarding "semantic alignment", we seek clarification regarding this metric. Could you please specify the quantitative metric you are referring to?
>
>
> **Q5: The paper completely omits standard practices for hyperbolic neural networks: No dimension reduction before hyperbolic operations (standard practice to reduce from C to C/r for computational efficiency [Khrulkov et al. 2020]:)**
>
> **A5**: After carefully reviewing the paper by Valentin Khrulkov et al. [ref3], we did not find the 'standard practice' of dimension reduction prior to hyperbolic operations being mentioned or implemented. There appears to be a discrepancy. Could this be due to a misinterpretation on our part or a different understanding on yours?
>
> [ref3] Hyperbolic Image Embeddings. CVPR20.
>
>
> **Q6: Eq. (10) notation error: g^W_L suggests window-Lorentz while g^P_E suggests patch-Euclidean, directly contradicting the text's claim of patch=Euclidean, window=hyperbolic**
>
> **A6**: We apologize for the error. It has been corrected in the revision as follows:
>
> > L284-285: Here, $g^W_\mathbb{L}$ represents features aggregated from windows via Hyperbolic similarity, while $g^P_\mathbb{E}$ does so for patches using Euclidean similarity.
>
> **Q7: Eq. (2) indexing bug: Uses c'_i but should be c'_j**
>
> **A7**: We thank the reviewer for pointing this out. Eq. 2 has been revised as follows:
>
> $$
> g_i = \sum _{i=1}^{M} \textbf{A} _{i,j} \cdot c_i^{\prime},~~~~\textit{s.t.}~~c _j^{\prime} = (c _j + \sum _{i=1}^{N} \textbf{A} _{i,j} \cdot x _i)/(1+\sum _{i=1}^{N} \textbf{A} _{i,j}).
> $$
>
> **Q8: No exploration of cluster numbers, window sizes, or curvature parameter κ**
>
> **A8**: Thanks for your suggestion. We agree that a comprehensive exploration of hyperparameters, such as the number of clusters, window sizes, and the curvature parameter κ, could potentially lead to further performance gains. However, given the extensive number of hyperparameters that could be tuned, **conducting comprehensive experiments on all these factors would require substantial computational resources and time**. Our current selection of hyperparameters, although empirically determined, has been validated through preliminary experiments to ensure that it establishes a robust and stable baseline. **The marginal performance gains from such explorations would not meaningfully strengthen our main claims, as the primary value lies in our framework design and its associated strategies.**
>
> Nevertheless, we acknowledge the reviewer's point and included experiments to investigate curvature sensitivity, as:
>
> | curvature  | top-1(%)$\uparrow$ | top-5(%)$\uparrow$ |
> | :--------- | :---- | :---- |
> | K=0.1      | 74.9  | 92.4  |
> | K=1.0      | 75.0  | 92.5  |
> | K=10.0     | 74.8  | 92.4  |
>
> It can be seen that the variation in curvature exhibits minimal impact. We hypothesize that this may be attributed to the fact that the current head dimension (24 for Tiny, 32 for Small and Large) is already sufficient to capture the relational structure within the embedding space across varying curvature regimes. As noted in [ref1], lower dimensions may exhibit greater sensitivity to curvature, as the combination of low dimensionality and high curvature excessively constricts the capacity of the embedding space.
>
> Furthermore, we have added a sentence in the Section D of the supplementary material (L866-870) stating that the performance of the model could potentially be further optimized through more extensive hyperparameter tuning, and we regard this as a valuable direction for future work.
>
> > L866-869: Besides, this paper does not conduct comprehensive investigations into hyperparameter configurations such as cluster numbers or window sizes. The model's performance could potentially be further optimized through systematic exploration of these hyperparameters, which warrant further investigation in future work.
>
> [ref1]: Hyperbolic Image Segmentation. CVPR22.

---

> ### Author Response · Authors · 2025-11-25
>
> **Q9: How is your method actually implementable? Please provide the exact dimensions at each step: C → ? (reduction?) → hyperbolic (C+1?) → ? (projection?) → concatenation → FC. Without these details, the method cannot be reproduced.**
>
> **A9**: Sorry for this misunderstanding.
> Taking Stage 1 of HCFormer-Tiny as an example, the input embeddings with 32 dimensions are first projected to 96 dimensions via a linear layer. These are then split into 4 heads, each with a dimension of 24. The dimension remains 24 throughout center estimation, cluster assignment, and token mixing, yielding an aggregated feature of dimension 24. These aggregated features are then concatenated across heads, restoring the dimension to 96. The process remains consistent for both patch-level and window-level clustering. Finally, the two aggregated features are combined, yielding a 192-dimensional representation, which is subsequently projected back to 32 dimensions through another linear layer.
>
> | Step  | Operation | Dimension |
> | :---- | :---- | :---- |
> | 1 | Project input embeddings via a linear layer | $32\rightarrow96$ |
> | 2 | Split features into 4 heads | $96\rightarrow4\times24$ |
> | 3 | Center estimation, cluster assignment, and token mixing | $24\rightarrow24$ |
> | 4 | Concatenate aggregated features from all heads | $4\times24\rightarrow96$ |
> | 5 | Combine aggregated features from both Patch-level & Window-level clustering | $96+96\rightarrow192$ |
> | 6 | Project the combined features via a linear layer | $192\rightarrow32$ |
>
> For comprehensive details, please refer to Table 8 in the Appendix, which provides the detailed model configurations. In addition, as indicated in Line 29, we commit to releasing the code upon acceptance of the paper.
>
>
> **Q10: What is the true parameter count? If FC reduces (2C+1) → C after concatenation, this adds significant parameters. Why isn't this in Table 1?**
>
> **A10**: The concern appears to stem from a misunderstanding. Indeed, the FC operation (2C $\rightarrow$ C reduction) defined in Eq. 10 does introduce additional parameters into the model. Therefore, as shown in Table 1, our model exhibits a certain increase in parameter count compared to CoC, FEC.  In addition, the learnable relative positional bias contributes as another source of parameter growth.
>
>
> **Q11: How do you handle high-dimensional hyperbolic operations? Processing 320-512 dims in hyperbolic space without dimension reduction contradicts standard practices. How is this numerically stable?**
>
> **A11**: As detailed in Table 5 of the supplementary material, HCFormer-Tiny maintains a head dimension of 24, while HCFormer-Small and HCFormer-Medium employ a head dimension of 32. These are considerably lower than the mentioned "320-512 dims".
>
>
> **Q12: Why not test Euclidean-Euclidean dual-path? This critical control would reveal whether improvements come from hyperbolic geometry or simply dual-path processing.**
>
> **A12**: In the first row of Table 4b and the third row of Table 4c, we present the Euclidean-Euclidean dual-path results, where "without hyperbolic geometry" indicates that all clustering operations are conducted within the Euclidean geometry. To enhance clarity, we have revised Table 4b as follows:
>
> | Windows | Patch | top-1(%)$\uparrow$ | top-5(%)$\uparrow$ |
> | :--- | :--- | :--- | :--- |
> | Euc. | Euc. | 73.4 | 91.4 |
> | Hyp. | Hyp. | 74.7 | 92.3 |
> | Hyp. | Euc. | 75.0 | 92.5 |

---

> ### Author Response · Authors · 2025-11-27
>
> Dear Reviewer,
>
> We sincerely appreciate the time and effort you have dedicated to reviewing our submission. We have submitted our rebuttal and would like to follow up to inquire whether our responses have sufficiently addressed your concerns.
>
> Please let us know if you have any remaining questions or require additional clarification. We value your feedback and are eager to ensure our work meets the highest standards.
>
> Authors

---

### Official Review · Reviewer_iogH · 2025-11-04

**Soundness:** 3
**Presentation:** 3
**Contribution:** 2
**Rating:** 6
**Confidence:** 4

**Summary:**

This paper introduces HCFormer, a novel vision backbone architecture that leverages interpretable clustering for token mixing, moving away from the black-box nature of convolution, attention, and MLP-based mixers. One of the main contributions is ClusterMixer, which explicitly mixes tokens via hierarchical clustering. To better capture the tree-like, hierarchical relationships in visual data, clustering is performed in hyperbolic space at the window level, while Euclidean space is used for fine-grained patch-level clustering. This dual-geometry approach enables HCFormer to efficiently and transparently aggregate information across both local and global contexts. The architecture achieves linear computational complexity, strong multi-task flexibility, and built-in interpretability. Extensive experiments show HCFormer consistently outperforms comparable models in image classification, and achieve decent results on semantic segmentation, object detection, and instance segmentation. However more results are comparison with CLIP and multi-object settings will make the paper stronger.

**Strengths:**

1) HCFormer replaces opaque token mixers (convolution, attention, MLP) with ClusterMixer, which uses explicit clustering algorithms for token mixing. This design provides transparency and interpretability, allowing users to understand how information is aggregated and propagated.
2) The model performs clustering in Euclidean space for local patch-level mixing and in hyperbolic space for global window-level mixing. This approach efficiently captures both fine-grained and abstract hierarchical relationships, improving semantic modeling and reducing distortion in feature aggregation.

3) By restricting clustering to local windows and using hierarchical strategies, HCFormer reduces the quadratic complexity of traditional clustering to linear, enabling efficient processing of high-resolution images and dense prediction tasks. The architecture is flexible and adapts well to various downstream tasks.
4)
HCFormer demonstrates strong results across multiple benchmarks, such as  ImageNet-1K, ADE20k, COCO etc.

**Weaknesses:**

1) Comparison to Hyperbolic CLIP[1]; it would be good to have CLIP based baselines in the paper, since it will show general applicability of the method.
2) How to extend this to multi-object settings? For example, When we have more than 1 object, how will this method extend? For example, On OpenImages as discussed in [2].
3) In line 299 it’s said “ Euclidean similarity is computed at the finer-grained and computationally demanding patch-level, while hyperbolic similarity is estimated at the abstract” ; what happens if both are done using hyperbolic loss instead of euclideian loss? It’s a known fact that hyperbolic models are hard to train, it would be good to understand why this particular setup was chosen.
4) On object detection the gains are very incremental, any particular reason why these gains are incremental when linear probing is actually decent?
Refrences
[1] Hyperbolic Image-Text Representations
[2] Hyperbolic Contrastive Learning for Visual Representations beyond Objects

**Questions:**

I think there are few more experiments that can help the paper specially in CLIP side. And some of the results seem incremental which needs to be properly discussed.

---

> ### Author Response · Authors · 2025-11-25
>
> We thank reviewer iogH for the valuable time and constructive feedback. We provide point-to-point response below.
>
> ----
>
> **Q1: Comparison to Hyperbolic CLIP[1]; it would be good to have CLIP based baselines in the paper, since it will show general applicability of the method.**
>
> **A1**: Per your request, we have incorporated additional CLIP-based baselines [ref1-3] in the revision:
>
> | Method            | top-1(%)$\uparrow$ |
> | :---------------- | :---- |
> | MERU ViT-S/16     | 34.3  |
> | MERU ViT-B/16     | 37.5  |
> | MERU ViT-L/16     | 38.8  |
> | HyCoCLIP ViT-S/16 | 41.7  |
> | HyCoCLIP ViT-B/16 | 45.8  |
> | HCL               | 58.5  |
>
> [ref1] Hyperbolic Image-Text Representations. ICML23.
>
> [ref2] Hyperbolic Contrastive Learning for Visual Representations beyond Objects. CVPR23.
>
> [ref3] Compositional entailment learning for hyperbolic vision-language models. ICLR24.
>
> **Q2: How to extend this to multi-object settings? For example, When we have more than 1 object, how will this method extend? For example, On OpenImages as discussed in [2].**
>
> **A2**: An interesting idea. A possible approach we currently envision involves constructing a hierarchical tree structure based on features of individual objects and their unions (which can be cropped from the feature map using corresponding bounding boxes), followed by implementing contrastive learning in hyperbolic space through parent-child and inter-child node interactions, similar to [ref4]. Once again, thank you for your valuable insights. This could certainly inform our future research directions.
>
> [ref4] LOGICSEG: Parsing Visual Semantics with Neural Logic Learning and Reasoning. ICCV23.
>
> **Q3: In line 299 it’s said “ Euclidean similarity is computed at the finer-grained and computationally demanding patch-level, while hyperbolic similarity is estimated at the abstract” ; what happens if both are done using hyperbolic loss instead of euclideian loss? It’s a known fact that hyperbolic models are hard to train, it would be good to understand why this particular setup was chosen.**
>
> **A3**: Following the reviewer's suggestion, we conducted further experiments using hyperbolic geometry at both the path and window levels. The results are summarized below:
>
> | Windows | Patch | top-1(%)$\uparrow$ | top-5(%)$\uparrow$ |
> | :--- | :--- | :--- | :--- |
> | Euc. | Euc. | 73.4 | 91.4 |
> | Hyp. | Hyp. | 74.7 | 92.3 |
> | Hyp. | Euc. | 75.0 | 92.5 |
>
> It can be observed that the clustering mixer relying entirely on hyperbolic geometry results in a slight performance decline and a significant reduction in computational efficiency (a decrease in throughput from 536.0 to 319.2).
>
> This discussion has been incorporated into the revision:
>
> > L460-462: When clustering is performed entirely in hyperbolic space, a marginal performance degradation (0.3% top-1) is observed, accompanied by a substantial reduction in computational efficiency (a decrease in throughput from 536.0 to 319.2).

---

> ### Author Response · Authors · 2025-11-25
>
> **Q4: On object detection the gains are very incremental, any particular reason why these gains are incremental when linear probing is actually decent?**
>
> **A4**: Thanks for your insightful suggestion.
> HCFormer-Small achieves solid performance gains over CoC and FEC, surpassing CoC-Small/49 by 1.5% in $\text{AP}^\text{box}$ and 0.6% in $\text{AP}^\text{mask}$, and outperforming FEC-Base by 0.8% in $\text{AP}^\text{box}$ and 0.5% in $\text{AP}^\text{mask}$, respectively.
> HCFormer-Medium still maintains a performance advantage over FEC-Large, with leads of 1.0% in $\text{AP}^\text{box}$ and 0.3% in $\text{AP}^\text{mask}$.
> However, its advantage narrows when compared to CoC-Medium/49.
> We posit that this is because object detection performance benefits from the number of cluster centers, which enables CoC-Medium/49 to achieve results competitive with ours.
> Crucially, this accuracy comes at the expense of efficiency; HCFormer-Medium achieves a 37% higher throughput (14.0 vs. 10.2) than CoC-Medium/49, underscoring its superior computational economy.
>
> This discussion has been incorporated into the revision:
>
> > L427-431: However, its advantage narrows when compared to CoC-Medium/49. We posit that this is because object detection performance benefits from the number of cluster centers, which enables CoC-Medium/49 to achieve results competitive with ours.
> Crucially, this accuracy comes at the expense of efficiency; HCFormer-Medium achieves a 37\% higher throughput (14.0 vs. 10.2) than CoC-Medium/49 (as shown in Supplementary Table 7), underscoring its superior computational economy.
>
> | Method | Param. (M) | GFLOPs (G) | Throughputs (images/s) | $\text{AP}^\text{box}\uparrow$ | $\text{AP}^\text{mask}\uparrow$ |
> | :- | :- | :- | :- | :- | :- |
> | CoC-Small/4  | 33.6 | 212.2 | 50.8 | 35.9 | 33.8 |
> | CoC-Small/25 | 33.6 | 212.2 | 35.3 | 37.5 | 35.4 |
> | CoC-Small/49 | 33.6 | 212.2 | 25.6 | 37.2 | 34.9 |
> | CoC-Medium/4 | 42.1 | 274.3 | 25.2 | 38.6 | 36.1 |
> | CoC-Medium/25| 42.1 | 274.3 | 15.2 | 40.1 | 37.4 |
> | CoC-Medium/49| 42.1 | 274.3 | 10.2 | 40.6 | 37.6 |
> | FEC-Small    | 24.3 | 178.8 | 31.2 | 35.6 | 33.6 |
> | FEC-Base     | 33.1 | 215.8 | 28.3 | 37.9 | 35.5 |
> | FEC-Large    | 47.1 | 277.9 | 13.6 | 39.9 | 37.3 |
> | HCFormer-Nano   | 24.0 | 173.5 | 54.9 | 36.0 | 34.0 |
> | HCFormer-Tiny   | 25.5 | 176.7 | 46.0 | 36.4 | 34.5 |
> | HCFormer-Small  | 34.1 | 216.0 | 33.0 | 38.7 | 36.0 |
> | HCFormer-Medium | 50.8 | 285.9 | 14.0 | 40.9 | 37.6 |

---

### Author Response · Authors · 2025-11-25

**Comment:**

We express our sincere gratitude to all reviewers for their valuable time and thorough assessment of our manuscript. We have revised our paper according to your comments. The major changes are as follows (the modifications are highlighted in red within the revision):

1. Following Reviewer fjaH's, UABQ's, and rvV3's suggestions, we have revised the manuscript to improve clarity.
2. In response to Reviewer iogH's comments, we have incorporated additional CLIP-based baselines in Table 1.
3. In response to Reviewer iogH's comments, we have conducted further experiments using hyperbolic geometry at both the path and window levels in Table 4b.
4. In response to Reviewer UABQ's comments, we have included more efficiency analysis in Section D of the supplementary material.
5. In response to Reviewer UABQ's comments, we have included the results of ablation studies on downstream tasks in Table 4c.
6. In response to Reviewer UABQ's comments, we have presented experimental results under varying curvature settings in Table 5a and Section C of the supplementary materials.
7. In response to Reviewer UABQ's comments, we have presented the results of varying the number of clusters (M) in the hyperbolic space in Table 5b and Section C of the supplementary materials.
8. In response to Reviewer UABQ's comments, we have re-measured the throughput of FEC and CoC on the identical hardware in Table 1.
9. In response to Reviewer rvV3's comments, we have added the more details about center estimation to Section B of the supplementary materials.
9. We have conducted additional experiments with a smaller-parameter variant (HCFormer-Nano) in Table 1,2,3.
10. We have modified the use of relative bias in downstream tasks (L293-294), which leads to improved performance and altered parameter counts in Table 2,3.

> L293-294: Additionally, the relative positional bias is linearly interpolated before use to accommodate variable input sizes.

Please refer to our response for more details. We have strived to address each of your concerns and welcome further discussions and insights.

Sincerely yours,

Authors.

---

### Meta-Review · Area_Chair_RM5P · 2025-12-21

**Summary:**

HCFormer replaces attention/MLP mixing with hierarchical clustering, using Euclidean patch-level clustering and hyperbolic window-level clustering for interpretability and efficiency; results are generally competitive with modest gains over CoC/FEC across ImageNet and downstream tasks.

Strength
* Clean, intuitive design; broad experiments and ablations.
* Rebuttal improved clarity (dims, controls, curvature/cluster sweeps) and added baselines.

Weakness
* One reviewer raised serious technical/clarity concerns (notation/implementation consistency), creating correctness risk.
* Novelty and gains are incremental; efficiency story mixed (classification throughput not clearly better).
* Interpretability claims remain under-quantified.

Overall, borderline contribution + lingering technical/confidence issues outweighed the steady-but-modest empirical improvements.

**Reviewer Scores:**

n/a

---

### Decision · Program_Chairs · 2026-01-26

Reject